# Exploring the First Aerosol Indirect Effect over Southeast Asia Using a Ten-Year Collocated MODIS, CALIOP, and Model Dataset

Alexa D. Ross[1], Robert E. Holz[1], Gregory Quinn[1], Jeffrey S. Reid[2], Peng Xian[2], F. Joseph Turk[3], Derek J. Posselt[3]

5  [1]Cooperative Institute for Meteorological Satellite Studies, University of Wisconsin - Madison, Madison, WI, 53706, USA
[2]United States Naval Research Laboratory, Monterey, CA, 93943, USA
[3]Jet Propulsion Laboratory, Pasadena, CA, 91009, USA

*Correspondence to*: Alexa D. Ross (alexa.ross@ssec.wisc.edu)

**Abstract.** Satellite observations and model simulations cannot, by themselves, give full insight into the complex relationships between aerosols and clouds. This is especially true over Southeast Asia, an area that is particularly sensitive to changes in precipitation, yet possesses some of the world's largest observability and predictability challenges. We present a new collocated dataset, the Curtain Cloud-Aerosol Regional A-Train dataset, or CCARA. CCARA includes collocated satellite observations from Aqua's Moderate-resolution Imaging Spectroradiometer (MODIS) and the Cloud-Aerosol Lidar with Orthogonal Polarization (CALIOP) with the Navy Aerosol Analysis and Prediction System (NAAPS). The CCARA dataset is designed with the capability to investigate aerosol-cloud relationships in regions with limited aerosol retrievals due to high cloud amounts by leveraging the NAAPS model reanalysis of aerosol concentration in these regions. This combined aerosol and cloud dataset provides coincident and vertically resolved cloud and aerosol observations for 2006-2016. Using the model reanalysis aerosol fields from the NAAPS and coincident cloud liquid effective radius retrievals from MODIS (cirrus contamination using CALIOP), we investigate the first aerosol indirect effect in Southeast Asia. We find that as expected, aerosol loading anti-correlates with cloud effective radius, with maximum sensitivity in cumulous mediocris clouds with heights in the 3-4.5 km level. The highest susceptibility in droplet effective radius to modeled perturbations in particle concentrations were found in the more remote and pristine regions of the western Pacific Ocean and Indian Ocean. Conversely, there was much less variability in cloud droplet size near emission sources over both land and water. We hypothesize this is suggestive of a high aerosol background already saturated with cloud condensation nuclei even during the relatively clean periods in contrast to the remote ocean regions which have periods where the aerosol concentrations are low enough to allow for larger droplet growth.

## 1 Introduction

Southeast Asia (hereby SEA) has long been identified as an area of climate vulnerability (e.g. Cruz et al., 2007; Hijioka et al., 2013; Lin et al., 2014). As outlined in the review Reid et al. (2013), the region has seen dramatic shifts in land use and anthropogenic emissions. While anthropogenic influences on the aerosol system are punctuated by dramatic deforestation and biomass burning events in both the northern insular SEA of Myanmar, Thailand, Vietnam, Cambodia and Laos, and the Maritime Continent of Indonesia, Malaysia, Singapore, Brunei and the Philippines, the region can also host pristine environments. Often thought of as remote, the population density in these regions can be exceptionally high, and the influence of industrial and domestic emissions is far reaching.

Large-scale climate change has manifested itself in temperature and rainfall signals (e.g., Cruz et al., 2013; Sharma and Babel, 2014). A persistent question in the scientific community is: to what extent do aerosol emissions feedback onto meteorology and perhaps climate? Are aerosol particles a primary contributor to regional climate change, or do aerosol emissions have second or third order influences? There is evidence that severe storms in the region are enhanced by aerosol emissions (Yuan et al., 2011) and the range of aerosol conditions in SEA theoretically should lead to changes in cloud physics (e.g., Sorooshian et al., 2010; Rosenfeld et al., 2014). However, while a straightforward question with theoretical and some observational

backing, tracing aerosol influences from emissions to climate impact is difficult in any environment. Addressing this question in Southeast Asia is particularly daunting. Complex monsoonal flows, organized convection, and near ubiquitous cirrus challenge observations and models alike (Reid et al., 2013).

NASA sponsored an Interdisciplinary Science (IDS) grant to systematically examine the sensitivity rainfall in SEA has to aerosol particle effects. But given the complexity and poor observability of the Southeast Asian system, proper and testable hypotheses needed to be developed. A reasonable insight was provided by Cruz et al. (2013), who found that there has been a steady increase in the number of "no-rain" days in Philippine ground stations. The increase in "no-rain" days occurs at the expense of "trace and light rain" days rather than moderate to heavy rainfall. Since it is trace or light rain events that appear to be largely affected, and such rain is often associated with precipitation from isolated warm convection that is thought to be more sensitive to increases in cloud condensation nuclei (CCN), perhaps this trend is associated with increasing aerosol loadings. Indeed, warm rain impacts have been repeatedly modeled (e.g., Langmann, 2007; Graf et al., 2009). However, aerosol particle lifecycle and regional meteorology are tightly coupled (e.g., Reid et al., 2012, 2015, 2016). Any study on SEA aerosol impact must then be wary of statistical confounding between aerosol and cloud phenomenon and any potential contextual, sampling, or measurement biases.

One approach to studying aerosol impacts in Southeast Asia is to make use of the NASA afternoon-train satellite orbit constellation (or A-Train). The A-train is anchored by Aqua's Moderate-resolution Imaging Spectroradiometer (MODIS) instrument, with significant synergy with other sensors such as the Cloud-Aerosol Lidar with Orthogonal Polarization (CALIOP) instrument on the Cloud-Aerosol Lidar and Infrared Pathfinder Satellite Observations (CALIPSO) spacecraft and the Cloud Profiling Radar (CPR) on the CloudSat spacecraft. Both MODIS and CALIOP instruments are sensitive to aerosol particles and clouds, but in regions with high cloud fraction such as SEA, there is often difficulty observing both cloud and aerosol particles in an atmospheric column. MODIS Collection 6 aerosol products, for example, have become quite clear-sky conservative, yielding few retrievals in even moderately cloudy environments. Optically thin cirrus fraction in SEA complicates aerosol and low to mid-level cloud retrievals alike (Chew et al., 2011; Reid et al., 2013). Certainly, profiling instruments like CALIOP lack the ability to observe aerosol particles below a cloud layer if the lidar's signal becomes completely attenuated by the cloud. CALIOP can only profile to an optical depth of 3 approximately. Thus, even error propagation and boundary layer aerosol retrievals are problematic after penetration through optically thin clouds.

For the IDS project, one of the approaches was to develop an A-train "curtain product" for SEA where these satellite products were physically collocated and combined with model meteorology and aerosol data. The model can provide information on variability of aerosol properties through the atmospheric column, including estimates of aerosol type and source location based on back-trajectory analysis, providing information about the aerosol properties in the presence of cloud. CALIOP can provide context on cloud top height and identify the presence of thin cirrus, while MODIS provides aerosol and cloud retrievals. This combined product provides a unique suite of data to investigate aerosol-cloud-precipitation relationships with the needed information to control for environmental or observational confounding. The use of the collocated NAAPS model reanalysis as a proxy for direct retrievals to define the aerosol concentration and type provides a new approach to investing the cloud-aerosol

interactions in regions such as SEA where the AOD retrievals are limited. This manuscript focuses on the relationship between aerosol amount and cloud droplet size using the combined dataset. As a continuation of this research, subsequent manuscripts will include analysis using CloudSat observations to investigate aerosol-precipitation relationships. Due to a battery anomaly in 2011, CloudSat left the A-Train orbit, so collocated CloudSat-MODIS-CALIOP observations only exist from 2006-2011.

Thus, subsequent research investigating aerosol-precipitation relationships will be limited to a five-year time period. For the research presented in this manuscript, we utilized the entire collocated MODIS-CALIOP record from 2006-2016.

The focus of this analysis is to investigate the cloud-aerosol relationship on cloud effective radius and the resulting impact on the cloud reflectance known as the first aerosol indirect effect (or FAIE), introduced by S. Twomey in 1974, with the premise that for aerosol particles to affect precipitation in isolated warm cumulus clouds, they likely have to first influence cloud

effective radius ($r_{eff}$). Using the curtain dataset described herein, we first explore the observational challenges of the aerosol and warm convective cloud system, followed by an analysis that includes the use of Navy Aerosol Analysis and Prediction System (or NAAPS; Lynch et al., 2016) reanalysis for aerosol detection. Indeed, model aerosol data has been combined with satellite retrieved properties before. Ma et al. (2010) uses aerosol concentrations simulated by the GOCART model to show a dependency of observed cloud droplet size – provided by MODIS – on organic carbon amounts. The study by Ma et al. (2010)

was for a global domain and did not investigate the long-reaching effects of aerosol particles or the scale of the impact of pollution on cloud droplet size. Additionally, Ma et al. (2010) did not investigate the dependency of the FAIE on cloud top height, which can be used as a proxy for filtering different types of clouds.

This paper focuses on the MODIS, CALIOP and NAAPS components of the Southeast Asian Curtain Cloud-Aerosol Regional A-Train (CCARA) dataset, and structured in the following way. In Sect. 2, we describe the generation of the A-Train curtain

dataset. In Sect. 3, we examine the baseline statistical properties and quality assurance of the products therein, with a particular emphasis on sampling and the potential for thin cirrus cloud contamination. Sect. 4 provides results and discussions on observed relationships between aerosol and cloud $r_{eff}$. A final discussion and conclusion synopsis is provided in Sect. 5.

## 2 Methods: Development of the Southeast Asian Curtain Cloud-Aerosol Regional A-Train (CCARA) product

The dataset used here was developed specifically to meet the challenges of doing aerosol and cloud research in the complex

SEA environment. Here, we demonstrate this system at its simplest level as a launching point for more detailed analysis of the CCARA. The dataset is comprised of data from two satellite instruments and fields from two models. In short, the basis is CALIOP providing cirrus screening and cloud height information. Other cloud properties are derived from MODIS. Aerosol properties are derived from the NAAPS-RA (which includes aerosol optical depth (AOD) from MODIS and the Multi-angle Imaging Spectroradiometer (MISR) instrument AOD), with supporting meteorological information from the ERA-Interim

atmospheric reanalysis (or European Reanalysis dataset) (Dee et al., 2011) produced by the European Centre for Medium-Range Weather Forecasts (ECMWF).

Sect. 2.1 includes details of the satellite instruments. Sect. 2.2 describes the model fields. Sect. 2.3 explains the technique by which the satellites and model fields were collocated to build CCARA.

### 2.1 Description of the satellite instruments

MODIS flies on board the Aqua satellite, which launched in 2002 (Salomonson et al., 2002). The imager observes reflectance at 36 wavelengths in the visible and infrared bands and spatial resolutions ranging from 250 meters to 1 km. The MODIS products used in this research are from Level 2 Collection 6 (C6), provided at spatial resolutions spanning 1 to 10 km$^2$ at nadir.

In this study, the primary MODIS data used are cloud property products (MYD06; Platnick et al., 2017), the cloud mask products (MYD35; Ackerman et al., 1998) and aerosol product (MYD04, Levy et al., 2005). MODIS collocation indices are found using the MYD03 geolocation files at 1 km spatial resolution using the collocation methodology described in (Nagle and Holz 2009). Partly cloudy (PCL) pixels are classified in the C6 MYD06 algorithm, but are not included in this analysis. The CALIOP instrument provides observations at two wavelengths: 532 nm (visible) and 1064 nm (near-infrared) with polarization capabilities in the 532 nm channel. The lidar has a vertical resolution of 30 m, producing detailed profiles of cloud and aerosol vertical structure. CALIOP Level 2 cloud products exist at horizontal resolutions of 333 m, 1 km, and 5 km. CALIOP Level 2 aerosol products exist at 5 km horizontal resolution. All of the CALIOP data used in this analysis are from the 5 km cloud and aerosol products (CPRO and APRO) which while having 60 m vertical resolution, can include spatial averages up to 80 km depending on the signal-to-noise ratio. Version 3 (V3) of CALIOP data are used in this analysis. CALIOP Level 2 algorithms provide their own set of feature layers that can distinguish up to 10 cloud and aerosol layers (CLAY and ALAY; Winker et al., 2009). To further take advantage of the CALIOP-resolved profile without having to save the full CALIOP profiles, customized layers defined to integrate with NAAPS vertical layers were designed specifically for this research and are discussed in Sect. 2.3.2. We use CALIOP aerosol products to verify the NAAPS reanalysis, and CALIOP cloud products to define cloud top heights and screen for thin cirrus.

### 2.2 Description of model products

The NAAPS reanalysis (Lynch et. al., 2016) is a decade-long (2003-2016) global 1x1 degree and 6-hourly aerosol reanalysis product, which was developed and validated at the Naval Research Laboratory. This reanalysis utilizes a modified version of NAAPS as its core and assimilates quality-controlled retrievals of AOD from MODIS on Terra and Aqua and MISR on Terra (Zhang et al., 2006; Hyer et al., 2011; Shi et al., 2014). NAAPS characterizes anthropogenic and biogenic fine (including sulfate, and primary and secondary organic aerosols), dust, biomass burning smoke and sea salt aerosols. Smoke from biomass burning is derived from near-real-time satellite-based thermal anomaly data to construct smoke source functions (Reid et al., 2009), with additional orbital corrections on MODIS-based emissions and regional tunings to mitigate missing fire detection resulting from daily variations in orbital coverage and cloud obscuration. Aerosol wet deposition in the tropics is driven by NOAA Climate Prediction Center (CPC) MORPHing (CMORPH) precipitation derived from satellite observations (Joyce et al., 2004) to correct model precipitation biases that ubiquitously exist in numerical models (Xian et al., 2009). The 3-dimensional aerosol fields are resolved vertically into 25 layers between the surface and about 70 mb following a sigma-pressure coordinate.

In addition to NAAPS, the CCARA dataset includes ERA-Interim atmospheric reanalysis. Meteorological fields such as pressure, temperature, specific and relative humidity, winds, and divergence are included with the CCARA product. Although not used specifically in this paper on simple effective radius sensitivity, the ERA-Interim reanalysis provides important information on the cloud regional environment. While NAAPS derives its meteorology from the Navy Global Environmental Model (NAVGEM), the most consistent meteorology in the region is from the ERA-Interim. While this may result in some mismatch between the ERA-Interim meteorology and the NAVGEM-driven aerosol components, the key parameters of interest here, including wind shear, water vapor, and temperature profiles, would not substantively impact aerosol-meteorology comparisons and thus the best available data are used.

## 2.3 Architecture of the collocated MODIS, CALIOP, and NAAPS dataset

The CCARA dataset was processed for the domain of Southeast Asia and, specifically $-15°N <$ latitude $< 25°N$ and $90°E <$ longitude $< 140°E$. This dataset spans from August 2006 to December 2016, encompassing over ten years of observations and reanalysis. To collocate, a single footprint is generated at the same size as MODIS native infrared and cloud product spatial resolution (1 km$^2$) and includes data from MODIS, CALIOP, NAAPS, and ERA-interim reanalysis. Satellites are first collocated, then the gridded model reanalysis is mapped onto the collocated satellite data. A ten-year dataset was processed starting with data from 2006 through 2016, which includes more than 29 million collocated footprints over SEA. CALIOP, NAAPS, and ERA-interim are vertically resolved and matched to the remote sensing observations as discussed in Sect. 2.3.1.

## 2.3.1 Collocation of remote sensing observations and models

The first step to building the collocated dataset is physically collocating the satellite observations, which for this analysis are MODIS and CALIOP with native resolutions of 1 km$^2$ and 333 m, respectively. The collocation is initiated using CALIOP's native Level 1 resolution of 333 m, then downscaled to match the CALIOP's Level 2 cloud and aerosol products whose resolution is 5 km. The physical collocation methodology assigns a single "master" instrument's observation with subsequent "follower" observations collocated into the master field of view (FOV) (Nagle and Holz, 2009). Typically, the master instrument has the largest surface FOV, which for this dataset is MODIS. The collocation software projects the master FOV (from MODIS) onto the Earth's surface, accounting for the instrument angular FOV and scan angle. Then the software identifies the follower FOVs (from CALIOP) that are coincident with the master observation. Up to four CALIOP 333 m FOVs are collocated within a 1 km MODIS footprint as illustrated in Fig. 1. Data from each of the CALIOP FOVs are averaged or reported as a fraction (e.g., cloud fraction) and assigned to a single MODIS footprint. The CALIOP data are weighted equally. Physically collocating the observations allows for direct intercomparison between the observations including the variability of follower pixels within the MODIS FOV (Nagle and Holz, 2009). We use this variability as a filter to reduce the uncertainty in our analysis as will be discussed in Sect. 2.4.

With the satellites collocated, the closest (in both space and time) NAAPS reanalysis 1-degree grid cell is identified with the model fields are added to each MODIS-CALIOP footprint in each file (or *granule)*. Each granule contains approximately 2000

points (the 'along-track' dimension of a MODIS granule). NAAPS and ERA-interim fields are inserted into existing MODIS-CALIOP FOVs simply by finding and assigning $1° × 1°$ grid fields to the appropriate MODIS-CALIOP FOV.

### 2.3.2 Vertical data

Collocating the CCARA dataset in the vertical dimension adds significant complexity. CALIOP Level 2 cloud and aerosol
layer data provides up to 10 layers defined by the CALIOP feature-finding algorithm (Winker et al., 2009). However, the layers' altitudes are dynamic and are not at a high enough vertical resolution to be directly compared to the NAAPS vertical aerosol distributions. In addition to the layer products, the CALIOP Level 2 cloud and aerosol products are provided as profiles with both the cloud and aerosol extinctions retrieved at 30 m vertical resolution. To facilitate direct comparison between the model fields and the CALIOP cloud and aerosol products, we have defined two sets of customized vertical layers for CCARA:
one that captures the full vertical column at 1 km vertical resolution, and one at slightly higher resolution that captures the planetary boundary layer (PBL).

The first set of layers that captures the full atmospheric column is defined with respect to the mean sea level (MSL). They parse the vertical dimension from 0-18 km in 1 km increments (each layer is 1 km in thickness). These are referred to as MSL layers. The second set of layers has a finer vertical resolution and is defined with respect to the surface elevation; these nine
layers are referred to as PBL layers. The nine PBL layers, defined with respect to surface elevation, are: 200-500 meters, 0.5-1 km, 1-1.5 km, 1.5-2 km, 2-2.5 km, 2.5-3 km, 3-4 km, 4-5 km, and 5-6 km. The two sets of layers are illustrated in Fig. 2. We do not include a 0-200 meters layer because the effects of surface reflectance on CALIOP retrievals (Winker et al., 2009). For each layer in the MSL and PBL layer sets, NAAPS extinctions and aerosol mass concentrations are interpolated and saved. These are separately attributed to four aerosol species: anthropogenic and biogenic fine mode (or ABF), dust, smoke, and sea
salt, and two aerosol modes: fine and coarse. The CALIOP aerosol and cloud extinction profiles are integrated within each layer providing the layer optical depth. The fractions of cloud or aerosol detected within the layer by CALIOP are also saved. These CALIOP-derived layer products are calculated using the V3 Level 2 profile products which provide aerosol and cloud extinctions at 30 m resolution. Vertically-resolved ERA-interim meteorology fields (the European Reanalysis Interim data) are also included for the PBL and MSL layers.

**2.4 Analytical techniques**

This section details the methods of analysis and describes the definition of the First Aerosol Indirect Effect (FAIE) using the CCARA dataset. While the dataset has many dimensions of aerosol, cloud and meteorological parameters, for this first introductory paper we are looking at the AOD-cloud comparisons, accounting for the regional biases present in nearly all datasets investigating Southeast Asia (Reid et al., 2013). Future papers will further explore aerosol-cloud indirect effects in
SEA.

*Input intercomparison:* This analysis aims to evaluate the extent to which aerosol particle concentrations and type (defined by NAAPS) impact the cloud properties (namely cloud effective radius) of liquid clouds. To this end, we cross-correlated modeled

aerosol optical depth anomalies (provided by NAAPS) with cloud $r_{eff}$ (observed by MODIS), using CALIOP to filter for cirrus contamination and provide the cloud top height. As an initial validation of this approach, we first intercompare the total columnar AOD between satellite observations and the NAAPS over land and ocean. AOD innovation is the observed AOD minus the model-derived AOD, or (AOD$_{obs}$-AOD$_{NAAPS}$), where the observation can be CALIOP or MODIS. For ocean

retrievals, AOD innovations average 0.02 for MODIS-NAAPS and -0.05 for CALIOP-NAAPS. Over land, MODIS-NAAPS innovation has an average of 0.05 compared to 0.07 for CALIOP-NAAPS.

*Partitioning the region:* SEA is partitioned into oceanic and land sub-regions as we hypothesize different regions of the domain will have varying sensitivities to the FAIE. Ocean grids and land regions are shown in Fig. 3. Ocean retrievals are divided into twenty 10x10-degree grids to analyse the FAIE (Fig. 3 right panel). To compute thresholds for ocean aerosol anomalies, we

use four larger ocean grids to reduce the uncertainty in the anomaly calculation (Fig. 3 left panel). Land retrievals are divided into six regions which are: 1) Peninsular Southeast Asia (Myanmar, Thailand, Cambodia, Laos, Vietnam), 2) Sumatra and peninsular Malaysia, 3) Borneo, 4) Java, 5) the Philippines, and 6) The eastern domain of Sulawesi, Bali, Timor, and New Guinea (hereby East Indonesia). Land aerosol anomaly thresholds are calculated using the same six land regions.

*Aerosol anomalies*: Like all global aerosol models, it is expected that the NAAPS model has varying regional efficacy. Yet,

global models can be expected to simulate large perturbations for large-scale aerosol features (e.g., Sessions et al., 2015), but finer scale local and mesoscale effects with poorer fidelity. Thus, direct comparisons between the global aerosol model and $r_{eff}$ can lead to regionally varying biases, which in turn can cloud the impact of varying aerosol concentrations on cloud properties. However, perturbations in the large-scale model should reflect regional changes associated with significant biomass burning or anthropogenic pollution events. Thus, correlating NAAPS aerosol perturbations to cloud properties can

isolate how these large events can influence the cloud environment. Anomalous aerosol amounts were defined by examining the full column fine mode AOD from NAAPS. A scene is determined to be anomalous if the AOD is more or less than an average AOD by one standard deviation. The AOD anomaly thresholds were calculated using the ten-year NAAPS AOD that have been mapped into the CCARA product. Distributions of aerosol mass loading were divided by species, geographic region, surface type, and season. For each ocean grid and land region, ten-year distributions of aerosol load (integrated aerosol mass

concentration) are found separating boreal winter and summer months (i.e. winter is December through May and summer is June through November).

*Aerosol thresholds:* Our analysis uses the NAAPS fine mode AOD anomalies. Fine mode AOD is used as opposed to total or speciated AOD to encompass any possible aerosol acting as CCN. Distributions of NAAPS AOD that are collocated into the CCARA dataset are compiled. A mean AOD ($\mu_{AOD}$) and standard deviation of AOD ($\sigma_{AOD}$) are calculated for each region and

season. We assume distributions of AOD are log-normal, thus $\mu_{AOD}$ and $\sigma_{AOD}$ are the geometric mean and geometric standard deviation of an AOD population. A pixel with NAAPS AOD $< (\mu_{AOD} \div \sigma_{AOD})$ is considered a negative anomalous aerosol event (clean event); AOD $> (\mu_{AOD} \times \sigma_{AOD})$ is a positive anomalous aerosol event (polluted event). This threshold is defined such that for each FOV in the CCARA dataset, there is an assignment of more/less/null aerosol amount for each of the four aerosol species and each of the two aerosol modes. Understandably, different regions experience different clean or polluted seasons

due to seasonal and regional patterns of biomass burning and land-use change (Xian et al., 2013; Reid et al., 2013). Thus, we define aerosol anomaly as relative to where and when data are, which for this manuscript are defined as a function of the regions in Fig. 3.

*Filtering*: Distributions of MODIS liquid $r_{eff}$ can be partitioned using a range of criteria such as region and surface type (land
or ocean), season, cloud top height, and aerosol anomaly. To identify correlations, bulk 'baseline' distributions are calculated that do not filter by any AOD threshold; baseline distributions are meant to serve as a benchmark or control to compare with aerosol anomalous distributions. Cloud top heights are defined using the CALIOP cloud layer product and are divided into three regimes which include layers between 1-3 km (small cumulus), 3-4.5 km (cumulus mediocris), and 4.5-6 km (smaller cumulus congestus), defined above surface elevation. Typically, at 6 km, temperatures are at approximately -5˚C, ensuring
that clouds are likely not experiencing ice physics. However, clouds at this temperature may be mixed in phase. We use a 6 km height cut-off and liquid-only MODIS retrievals to increase the certainty that we are looking only at liquid water $r_{eff}$ retrievals. CALIOP Level 2 cloud layer products are also used to filter FOV with thin ice cirrus contamination above the liquid cloud layer (Winker et al., 2009), which are removed from the analysis. From these filtered distributions, averages (geometric means) of $r_{eff}$ for baseline, polluted (positive aerosol anomaly), and clean (negative aerosol anomaly) scenarios are calculated.
Our goal is to investigate the FAIE by comparing the MODIS $r_{eff}$ for polluted versus clean data to the baseline mean $r_{eff}$. The polluted/clean displacement of $r_{eff}$ from the baseline $r_{eff}$ is called the FAIE signal.

*Sampling bias and statistical significance:* CALIOP is a nadir-viewing instrument in a polar sun-synchronous orbit. Even with a ten-year record, the sampling statistics when accumulated over small regions for a subset of clouds can be poor. For example, summertime data over the Philippines for land surfaces and clouds existing between 1-3 km yield only 1005 collocated MODIS
and CALIOP $r_{eff}$ retrievals. This limited sampling results not just from the nadir viewing geometry, but also from the extensive cirrus cloud cover in the region that attenuates the CALIOP observations and limits the low-level cloud MODIS retrieval yield. After applying an additional filter for investigating the aerosol anomaly, only 40 of those retrievals occur during a positive fine mode aerosol anomaly. For comparison, the 1-3 km summertime clouds in the northernmost and easternmost 1-degree ocean grid have 8603 baseline $r_{eff}$ and 461 retrievals after filtering for positive fine mode aerosol anomaly due to the larger
surface area of this region and much higher frequency of low clouds over this ocean grid compared to over the Philippines. Statistical significance was determined by performing a two-sample t-test on each partitioned distribution of MODIS $r_{eff}$. The average clean and polluted $r_{eff}$ are compared to the baseline $r_{eff}$. p-values are calculated and interpreted to give the certainty that the mean of an anomalous distribution is significantly different from the mean of a baseline distribution. We also specify the type of alternative hypothesis that is tested in the t-test, meaning we can determine to what certainty the mean of an
anomalous distribution is more or less than the mean of a baseline distribution. Thus, we can answer specifically whether smaller cloud droplets are more likely observed in a polluted environment compared to the baseline (or "normal") environments. Similarly, we test the likelihood that larger mean $r_{eff}$ occur in a clean environment compared to the normal environments. We consider any p-value less than 0.01 to signify a significant difference between polluted/clean and baseline distributions.

## 3 Results

This section focuses on results from the ten-year CCARA dataset over Southeast Asia that includes observations from MODIS and CALIOP as well as NAAPS and ERA-interim reanalysis. Sect. 3.1 explores the basic properties of the CCARA dataset, including intercomparisons between NAAPS aerosol reanalysis and the CALIOP and MODIS aerosol observational data.

Then, we investigate the aerosol anomaly thresholds for each region and season, defined using the NAAPS fine mode AOD. We also show the overall mean (or baseline) MODIS effective radius ($r_{eff}$) for each region and season without considering aerosol anomalies.

Sect. 3.2 investigates the FAIE signal. To detect FAIE, we compare populations of baseline $r_{eff}$ to clean and polluted $r_{eff}$. Statistical significance testing is performed on these populations to determine whether the mean of clean or polluted $r_{eff}$ is

greater or less than the mean of the baseline $r_{eff}$ within degree of certainty. To pinpoint when and where the FAIE occurs, populations of $r_{eff}$ are filtered by region (twenty ocean grids and six land regions) and season (winter or summer). Ocean and land regions are shown in Fig. 3. To further narrow the occurrence of FAIE, we filter by other fields in the CCARA dataset such as CALIOP cloud top height.

### 3.1 Basic data properties

This section details the intercomparison of the NAAPS, MODIS, and CALIOP AOD in the ten-year CCARA dataset over Southeast Asia. Here we explore AOD intercomparisons between NAAPS reanalysis and CALIOP and MODIS aerosol observations. We also explore average AOD in each region and season, which we use to define aerosol anomaly thresholds. Lastly, we examine the mean effective radius in each region and season, which we use later in Sect. 3.2 to explore the FAIE.

### 3.1.1 Aerosol detection intercomparison: NAAPS versus the observations

An AOD intercomparison between NAAPS and the observing platforms CALIOP and MODIS is presented in Fig. 4, presenting 2-D histograms of $AOD_{MODIS}$-$AOD_{NAAPS}$ and $AOD_{CALIOP}$-$AOD_{NAAPS}$ (aerosol innovations) for the SEA domain with separate distributions for land and ocean retrievals. Recall that the innovation is simply the observed AOD minus the AOD derived from NAAPS reanalysis. Only retrievals where MODIS, CALIOP, and NAAPS all detect non-zero AOD are included. CALIOP was used to filter cirrus contamination and only include cloud-cleared AOD scenes.

Ocean AOD innovations average 0.02 for MODIS-NAAPS and -0.05 for CALIOP-NAAPS. Over ocean, the differences (MODIS–CALIOP) between the observed AOD values average 0.08, much larger than the differences when compared with NAAPS, indicating that the NAAPS bias is small relative to the direct observations. Over land, MODIS-NAAPS innovation has a mean of 0.05 compared to 0.07 for CALIOP-NAAPS. Land innovations have a wider distribution than ocean innovations for both MODIS-NAAPS and CALIOP-NAAPS. Over land, the MODIS-NAAPS innovations have a variance of 0.04 while

the CALIOP-NAAPS innovations have a variance of 0.2. Over ocean, these variances are 0.005 and 0.02 for MODIS-NAAPS and CALIOP-NAAPS, respectively.

The innovations for MODIS-NAAPS are smaller than those for CALIOP-NAAPS. Campbell et al. (2012) states that CALIOP is known to have a low bias. Overall, the AOD biases are small, with NAAPS in good agreement with the MODIS and CALIOP
observations. This provides confidence in the use of the model reanalysis to define the aerosol properties. NAAPS is understandably in slightly better agreement with MODIS, as the MODIS AOD is assimilated into the NAAPS reanalysis. However, MODIS assimilation occurs less frequently in SEA than in other regions due to persistent cirrus cloud cover. For CALIOP, the retrieval requires assumptions regarding the aerosol lidar ratio which can introduce systematic biases (Campbell et al., 2012).

The CCARA dataset provides vertically-distributed NAAPS and CALIOP aerosol extinctions, as discussed in Sect. 2.3.2. We compare layer extinctions in Fig. 5, which present biases and absolute values of aerosol extinction for PBL and MSL layers. For each layer, mean aerosol extinction and mean extinction bias are shown. The bias is the observed layer extinction minus the NAAPS reanalysis layer extinction.

The top panels of Fig. 5 show the absolute extinctions (left) and extinction biases (right) for NAAPS and CALIOP MSL layers.
Bias is defined as model extinction minus CALIOP observed extinction. Over both ocean and land, NAAPS overestimates CALIOP extinction at higher altitudes, and underestimates CALIOP extinction at lower altitudes. This is consistent with the AOD bias results in Fig. 4. The extinction bias switches from positive to negative at 3.5 km over land and 1.5 km over ocean. The bottom panels of Fig. 5 show extinctions and extinction biases for the PBL layers, which are a finer vertical resolution near the surface compared to MSL layers. The extinction biases are exaggerated for the finer vertical bins, demonstrating
NAAPS's underestimation of CALIOP layer bias over land. The PBL layer closest to the surface reports an extinction bias < -0.07 km$^{-1}$ for land retrievals. In summary, we consider NAAPS aerosol reanalysis in reasonable agreement with MODIS and CALIOP observations. However, NAAPS does demonstrate lower aerosol particle concentrations in the lower free troposphere compared to CALIOP.

### 3.1.2 Aerosol anomaly threshold

Aerosol anomalies are defined by calculating the geometric mean ($\mu_{AOD}$) and geometric standard deviation ($\sigma_{AOD}$) of NAAPS fine mode aerosol optical thickness for each sub-region and each season (boreal summer (JJASON) and boreal winter (DJFMAM)) during all ten years of NAAPS reanalysis collocated into the CCARA dataset. The NAAPS fine mode AOD can be thought of as encompassing AOD contributions from both anthropogenic and smoke aerosol species. NAAPS AOD are typically log-normally distributed (e.g., Lynch et al., 2016), therefore geometric $\mu_{AOD}$ and geometric $\sigma_{AOD}$ are calculated.
We define a positive aerosol anomaly, or polluted event, when the AOD is greater than an upper threshold, ($\mu_{AOD} \times \sigma_{AOD}$). A negative aerosol anomaly, or clean event, is when the AOD is less than a lower threshold, ($\mu_{AOD} \div \sigma_{AOD}$). Winter and summer upper and lower AOD thresholds for each region are shown in Table 1, along with the mean AOD values. Recall that to define

ocean aerosol anomalies, the region is split into four large grids, which we will distinguish by referring to as southwest, northwest, southeast, and northeast ocean grids as presented in Fig. 3. Note the seasonal differences in each region and recall that Indochina experiences a biomass burning season during winter months while Java, Sumatra, and Borneo experience a biomass burning season during summer months. These burning seasons are reflected in our AOD thresholds. Summer upper AOD thresholds for Java, Borneo, and Sumatra are much greater than their winter upper thresholds. Over Indochina, the winter upper AOD threshold over double that in the summer.  Mean AOD values for each region and season are mapped in Fig. 6.

| | Summer $\mu_{AOD} \div \sigma_{AOD}$ | Summer $\mathbf{\mu_{AOD}}$ | Summer $\mu_{AOD} \times \sigma_{AOD}$ | Winter $\mu_{AOD} \div \sigma_{AOD}$ | Winter $\mathbf{\mu_{AOD}}$ | Winter $\mu_{AOD} \times \sigma_{AOD}$ |
|---|---|---|---|---|---|---|
| Land - Indochina | 0.06 | **0.12** | 0.26 | 0.10 | **0.23** | 0.57 |
| Land - Philippines | 0.04 | **0.08** | 0.17 | 0.03 | **0.06** | 0.13 |
| Land - E. Indonesia | 0.04 | **0.08** | 0.15 | 0.03 | **0.06** | 0.11 |
| Land - Java | 0.11 | **0.19** | 0.32 | 0.08 | **0.15** | 0.28 |
| Land - Borneo | 0.06 | **0.13** | 0.30 | 0.05 | **0.09** | 0.15 |
| Land - Sumatra | 0.06 | **0.15** | 0.34 | 0.07 | **0.13** | 0.24 |
| Ocean - northwest | 0.04 | **0.09** | 0.20 | 0.07 | **0.16** | 0.33 |
| Ocean - northeast | 0.02 | **0.06** | 0.15 | 0.02 | **0.06** | 0.17 |
| Ocean - southeast | 0.03 | **0.07** | 0.15 | 0.02 | **0.05** | 0.11 |
| Ocean - southwest | 0.03 | **0.07** | 0.18 | 0.02 | **0.05** | 0.11 |

**Table 1:** Lower fine mode AOD thresholds, median AOD, and upper AOD thresholds, shown for winter and summer for each of the six land regions and four ocean grids. Thresholds were found by computing the geometric mean ($\mu_{AOD}$) and geometric standard deviation ($\sigma_{AOD}$). The lower threshold is $\mu_{AOD} \div \sigma_{AOD}$, any AOD below this is considered *clean*. The upper threshold is $\mu_{AOD} \times \sigma_{AOD}$, any AOD above this is *polluted*.

### 3.1.3 Baseline MODIS effective radius

In this section, we show the average (geometric mean or $\mu_g$) MODIS liquid effective radius for each region and season. We also filter effective radius by three cloud height regimes, which are introduced in Sect. 2.4. These regimes are cloud top heights between 1-3 km, 3-4.5 km, and 4.5-6 km.

An effective radius population with no aerosol anomaly filtering is referred to as a 'baseline' population. Populations of baseline $r_{eff}$ are compiled for each region and season for the ten-year dataset and the geometric mean of $r_{eff}$ is computed. MODIS liquid $r_{eff}$ retrievals are included only if CALIOP detects no cirrus above the liquid cloud top. Pixels classified as partly cloudy (PCL) by the MODIS cloud mask algorithm (MOD35) are not included to reduce uncertainty in the effective radius results. Maps of baseline geometric mean $r_{eff}$ are shown in Fig. 7. These maps are shown for each season and cloud height regime. In each map, three black bolded numbers accompany each region. The first signifies the population count and the second preceded

by an $s$ is the geometric standard deviation of the baseline $r_{eff}$ population. The $s$-number represents the "spread" of the baseline $r_{eff}$ population. The third number is the preceded by an $m$ is the median $r_{eff}$. As expected, average ocean $r_{eff}$ values are larger than land values as oceans regions are typically cleaner environments. Further, $r_{eff}$ is generally larger on the fringes of the domain, compared to $r_{eff}$ in the higher populated areas. This is especially true for the ocean grids over the West Pacific Ocean

in the north-eastern-most region of the SEA domain (see also Fig. 8c), where average $r_{eff}$ commonly exceeds 18 microns, compared to 8-10 microns over densely populated peninsular SEA and Java. Average $r_{eff}$ values also increase with increasing height, likely resulting from the growth of droplets in deeper clouds. Evidence of major burning seasons in Fig. 7 is seen over Indochina where the mean $r_{eff}$ for all cloud heights is much smaller during burning active winter compared to summer months. Similar small $r_{eff}$ over Java is seen during summer months. These maps suggest signs of cloud-aerosol interactions related to

the FAIE, where areas that are known to be polluted/clean experience smaller/larger effective radius retrievals. We now investigate if the mean $r_{eff}$ values are statistically smaller/larger during extreme polluted/clean events defined by NAAPS.

## 3.2 FAIE occurrence

Populations of MODIS liquid $r_{eff}$ are compiled from the CCARA dataset filtering each region and season for the ten-year record. We can filter the populations by aerosol thresholds using NAAPS aerosol reanalysis, allowing us to compare clean and

polluted $r_{eff}$ populations to baseline/background $r_{eff}$. From this analysis, we establish significant evidence for the FAIE as presented in Fig. 9 and Fig. 10. Significant FAIE is detected if the two-sample t-test provides a p-value < 0.01.

Distributions of these populations are shown in Sect. 3.2.1, where clean and polluted $r_{eff}$ are compared to baseline $r_{eff}$. In Sect. 3.2.2, we map the mean values of these distributions and investigate whether differences in distributions have statistical significance. Two-sample t-tests are conducted on the data to determine numerically whether means of clean/polluted $r_{eff}$ are

significantly larger/smaller than the baseline $r_{eff}$.

It is important to distinguish the *direction* of the FAIE. A *polluted FAIE* signifies that $r_{eff}$ is smaller relative to the region's baseline $r_{eff}$. A *clean FAIE* signal signifies that $r_{eff}$ is larger than the baseline $r_{eff}$.

### 3.2.1 Populations of $r_{eff}$ for baseline, polluted, and clean cases

Distributions of MODIS retrieved $r_{eff}$ were compiled separating each region and season, and cloud height regime. For

conciseness, select examples of the water cloud $r_{eff}$ distributions are shown in Fig. 8. The figure shows normalized distributions of baseline, polluted, and clean $r_{eff}$ of clouds between 1-3 km for six land regions: Indochina and the Philippines in winter and Borneo, Java, East Indonesia, and Sumatra in summer and as highlighted by the coloured regions in the lower right of Fig. 8. Distributions for the West Pacific and Indian Oceans are shown as well. The black lines show the baseline distributions of $r_{eff}$, i.e. $r_{eff}$ without a detected aerosol anomaly. Note that the baseline distributions vary widely in shape between each region. Figs.

8a (Indochina winter) and 8e (Java summer) have baseline $r_{eff}$ with a very small mode and small variance compared to Figs. 8b, 8c, 8d, 8f, and 8h, whose baselines are more widely distributed.

During boreal winter, drier conditions and enhanced biomass burning are present in the northern half of our study domain. Indochina distributions of $r_{eff}$ are very similar between baseline, polluted, and clean retrievals (Fig. 8a) with the peak of the distributions having small water cloud $r_{eff}$. This suggests that changes in aerosol concentration are not influencing significant change in the effective radius retrievals for the region, even during the burning season. Even during the relatively low aerosol concentration periods, there is significant nuclei to inhibit droplet growth. This region has high levels of biofuel, agricultural waste and industrial emissions (e.g., Reid et al., 2013), resulting in abundant sources of cloud nuclei with background conditions that may already saturate the CCN population. In comparison, the Philippines distributions (Fig. 8b) are different between the baseline, polluted, and clean retrievals during winter months. Polluted Philippines retrievals are smaller than the baseline; clean Philippines retrievals are larger than the baseline in contrast to Indochina. Based on Fig. 8, the Philippines are more sensitive to FAIE during winter months with the fine mode aerosol concentration impacting the MODIS effective radius retrievals over this region. Finally, the most dramatic sensitivity can be seen for the West Pacific boreal winter (Fig. 8c), the most remote of the regions examined.

During boreal summer, the monsoon flips to more wet conditions in the north and drier, more fire-prone conditions in the south. In Fig. 8d, Borneo behaves as would be expected owing to its relatively high population, little sensitivity in the baseline and more polluted conditions, but with some residual sensitivity for the "cleanest" of conditions. Such cleaner conditions do not exist on the highly populated island of Java, with considerably less sensitivity perturbations in AOD (see also Fig. 9). Similar findings were found for Sumatra (Fig. 8g). Some of the largest spreads in $r_{eff}$ for boreal summer were found in the more remote areas, such as Eastern Indonesia over land (Fig. 8f) or the Indian ocean – a frequent receptor for pollution from Java (Fig. 8h). Similar to the West Pacific winter in Fig. 8c, the Indian ocean experiences sensitivity during scenes of anomalous aerosol.

While the examples in Fig. 8 hint at signals of clean and polluted FAIE signals, we must consider the statistical significance of these distributions in the comparisons. The next section will systematically explore clean and polluted FAIE signals for each region in SEA and investigate signals of the FAIE where statistical testing has deemed the FAIE significant.

### 3.2.2 Mapped FAIE for winter and summer

In this section, we compute geometric means of $r_{eff}$ for baseline, polluted, and clean retrievals for each region, season, and cloud height regime. We use a t-test to determine whether the geometric means of those distributions are significantly different from one another to yield a FAIE signal. The significance test is designed such that we can test specifically if the clean $r_{eff}$ is *larger* than the baseline $r_{eff}$ (a clean FAIE) and if the polluted $r_{eff}$ is *smaller* than the baseline $r_{eff}$ (polluted FAIE).

The magnitude of the FAIE is the displacement of the anomalous (clean or polluted) $r_{eff}$ from the baseline $r_{eff}$:

$$\text{clean FAIE} = \text{clean } r_{eff} - \text{baseline } r_{eff} \geq 0$$
$$\text{polluted FAIE} = \text{polluted } r_{eff} - \text{baseline } r_{eff} \leq 0$$

This definition is defined so the clean FAIE is positive in sign and the polluted FAIE is negative. Fig. 9 shows maps of average $r_{eff}$ values for baseline populations, as well as maps of polluted and clean FAIE signals during summer months. Winter months are shown in Fig. 10. The rows in this figure signify different cloud top height regimes; the columns show the baseline $r_{eff}$, polluted FAIE signal, and clean FAIE signal. The cloud height regimes signified by the rows in Figs. 9 and 10 correlate from bottom to top: 1-3 km, 3-4.5 km, and 4.5-6 km. The FAIE signal maps are only shaded if the FAIE is deemed statistically significant ($p < 0.01$). Note that the same region may yield a significant clean FAIE, but not a significant polluted FAIE, or vice versa.

During winter months shown in Fig. 10, for clouds nearest to the surface (between 1-3 km) there is a significant clean and polluted FAIE (up to 6 microns in magnitude) for the West Pacific Ocean region, just east of the Philippines. This suggests that under heavily polluted environments, the average effective radius is up to 6 microns smaller relative to the effective radius under every-day scenes. Under extremely clean scenes, the average effective radius can be over 5 microns larger than the bulk average. Even for high clouds at 4.5-6 km, a polluted and clean FAIE signal is found for some of the West Pacific grids. A polluted and clean FAIE is also present for the Indian Ocean during winter (Fig. 10) and summer (Fig. 9). It appears maritime regions such as the West Pacific and Indian Oceans are some of the most susceptible to changes in $r_{eff}$ during anomalous aerosol amounts.

A very weak (< 1 micron in magnitude) clean and polluted FAIE signal exists for low clouds over Indochina. These FAIE signals are weak as the environment is likely almost always saturated with fine mode aerosol, resulting in additional sources having limited impact on the water cloud droplet size and the already low mean $r_{eff}$ shown in Fig. 9. Comparing Indochina in Fig. 9 to Fig. 8a, note that while the normalized histograms of clean and polluted $r_{eff}$ are almost identical, the slight difference in the right-side tails of this distribution, in addition to the high number of counts, yields a p-value small enough to consider this difference significant. The example of wintertime Indochina is consistent with the notion that there is a physical upper limit to cloud-aerosol interactions (Painemal and Zuidema, 2013). A slightly stronger FAIE over Indochina does appear for higher altitude clouds, though the average $r_{eff}$ for all (baseline) clouds over Indochina stays quite small and does not exceed 12 microns even for the tallest clouds.

The black bolded numbers in Fig. 9 and Fig. 10 signify the count of the cloud populations for each region in. The purpose of showing retrieval counts is to indicate whether the lack of a FAIE signal is due to low cloud counts or an insignificant change in the average anomalous $r_{eff}$.

Over some West Pacific peripheral ocean regions in Fig. 10, there is a lack of polluted liquid $r_{eff}$ retrievals for higher clouds (3-6 km). One reason for this is that the region is regularly covered in cirrus, reducing the likelihood of a liquid retrieval from MODIS. Additionally, while significant smoke and pollution can reach this region, it is somewhat rare and compounded by how we define aerosol anomaly. Our aerosol thresholds were defined by determining whether a certain AOD amount was anomalous relative to all clouds in that region during that season, *not* answering whether the AOD was anomalous to specific cloud top heights in that region during that season. Further, recall that ocean aerosol anomalies are determined for larger ocean grids, not the small 10x10-deg grids in Fig. 9 and Fig. 10. For these reasons, it follows that some easternmost West Pacific

polluted FAIE grids in Fig. 10 indicate a "0" count for taller cloud height regimes because any anomalously polluted clouds in that region are mostly likely to exist in the lowest regime closest to the surface.

During summer months shown in Fig. 9, a strong polluted FAIE appears across the SEA domain. Significant clean FAIE signals are present as well. Peripheral ocean regions in the West Pacific (upper right in maps) and Indian Ocean (lower left in maps) experience a strong polluted FAIE, with clouds decreasing in average $r_{eff}$ up to 6 microns for all cloud height regimes. It is clear that the burning season of Sumatra, Java, and Borneo has a great effect on cloud droplet size over ocean regions, with far-reaching effects into the West Pacific. It is likely that the nearly constant presence of anthropogenic aerosols from Indochina have an effect on West Pacific effective radius retrievals as well. The effect is emphasized during winter, which is Indochina's biomass burning season.

There is a complete absence of polluted FAIE signals over Java, Sumatra, and Borneo for 1-3 km clouds. These land regions have large populations, significant biofuel usage, and undergo burning seasons during these months. Java yields no polluted FAIE signal for *any* cloud heights. The mean $r_{eff}$ values in baseline and clean cases over Java are relatively low at 13 microns, implying that these regions are saturated with aerosol particles. We infer that relative changes in the aerosol loading having minimal effect on cloud droplet sizes for low clouds over regions of biomass burning.

**4 Discussion and conclusions**

Southeast Asia has long been thought of as particularly vulnerable to climate change. High population density and proximity to remote ocean regions highlights the complexity of observing aerosol indirect effects in this region. In this analysis, we have shown that the effects of anthropogenic and smoke aerosols on cloud properties are detectable using the ten-year collocated Curtain Cloud-Aerosol Regional A-Train (CCARA) dataset, which includes data from MODIS and CALIOP and the aerosol properties defined using model reanalysis from NAAPS. The use of the model instead of direct retrievals provides the means to define the aerosol properties despite the high frequency of cloud cover, which greatly limits the utility of the MODIS AOD retrievals in this region. Using the reanalysis from NAAPS paired with concurrent satellite cloud observations offers significant signs of a first aerosol indirect effect or FAIE. These are demonstrated throughout SEA, however there are significant trends in this effect which are correlated with both land and ocean as well by geographic region. The FAIE is defined when the mean effective radius is larger in anomalously clean cases and smaller in anomalously polluted cases compared to the overall (baseline) average $r_{eff}$. We call these effects the clean FAIE and polluted FAIE signals, respectively. FAIE signals are significant only if the $r_{eff}$ populations of anomalous and clean scenes pass a significance t-test and yield p-values < 0.01.

Sect. 3.2.2 shows that clean and polluted FAIE signals are quite common throughout the SEA domain during both winter and summer. Disregarding $r_{eff}$ in scenes of aerosol anomaly, baseline average $r_{eff}$ can vary greatly between regions. Biomass burning seasons are reflected in the baseline $r_{eff}$, as distributions of $r_{eff}$ are often condensed to very small values over land, as shown in examples in Fig. 8. Furthermore, burning seasons cause saturation in CCN for low clouds (1-3 km) over land. In these regions, changes in pollution have a limited impact on observed effective radius.

Smoke burning and anthropogenic aerosols have long-reaching effects on cloud properties in SEA, particularly in maritime regions. Figs. 9 and 10 show that peripheral ocean regions are often the most affected by influxes of anthropogenic aerosol; this is seen in winter and summer months. An important finding is that relatively clean ocean regions are particularly sensitive to anomalous AOD based on biomass burning seasons, showing the greatest variability (and sensitivity) in areas that are
thought of as remote or pristine.

There is often a weak or non-existent FAIE over regions that are saturated with aerosol, i.e. low clouds (cloud top height 1-3 km) over polluted land areas undergoing a biomass burning season. This is seen for Sumatra, Java, and Borneo during summer months and for Indochina during winter months where the baseline water cloud $r_{eff}$ is almost always low. This result signifies that under high pollution events in already polluted regions, changes in aerosol amount have little to no effect on cloud
microphysics. These same regions do sometimes yield a polluted FAIE for high liquid water clouds (cloud top height 4.5-6 km), signifying that the FAIE has some sensitivity to cloud height, possibly due the lower baseline aerosol concentration at the high levels.

Our results show significant and consistent signs of anti-correlation between observed liquid water droplet size (from MODIS $r_{eff}$) and aerosol amount (from NAAPS aerosol reanalysis). Smaller effective radius retrievals are associated with a surplus of
aerosol (polluted events) while larger effective radius retrievals are associated with a lack of aerosol (clean scenarios). These correlations depend on location and surface type, time of year, and cloud top altitude. An important finding from this analysis is that anthropogenic aerosol emissions in SEA have far-reaching impacts on cloud properties, affecting even high clouds in remote ocean regions that are often thought of as pristine. A clean and polluted FAIE is common throughout the domain. The FAIE appears strongest in peripheral ocean regions such as the south Indian Ocean and western Pacific Ocean, east of the
Philippines. We have concluded that the utilization of a ten-year collocated dataset that combines satellite cloud observations with modeling fields is a unique method to obtain statistically significant evidence of the FAIE in regions that are predominantly cloudy. We hope this study will serve as a precursor to investigating the indirect effects of aerosol on precipitation, or the second aerosol indirect effect (SAIE) with the addition of CloudSat as part of this analysis. We draw the connection between the FAIE and SAIE under the premise that in order for aerosol to affect precipitation processes in cumulus
clouds, they first have to affect cloud droplet size.

**Author contribution: The conception and design of this work was done by Jeffrey S. Reid and Robert E. Holz. Data collection was done by Alexa D. Ross and Peng Xian. Building and analysing the data was also done by Ross and Xian. Processing collocation code in order to build the dataset was done by Gregory Quinn. Interpretation of the data was**

done by Ross, Reid, Holz, Xian, F. Joseph Turk, and Derek J. Posselt. Drafting the article was done by Alexa Ross and Jeffrey Reid, with critical revisions by Holz, Xian, and Posselt. Final approval for publication was given by all authors.

Acknowledgements: Funding for this research was provided by the NASA Interdisciplinary Science Program, award number NNX14AG67G.

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

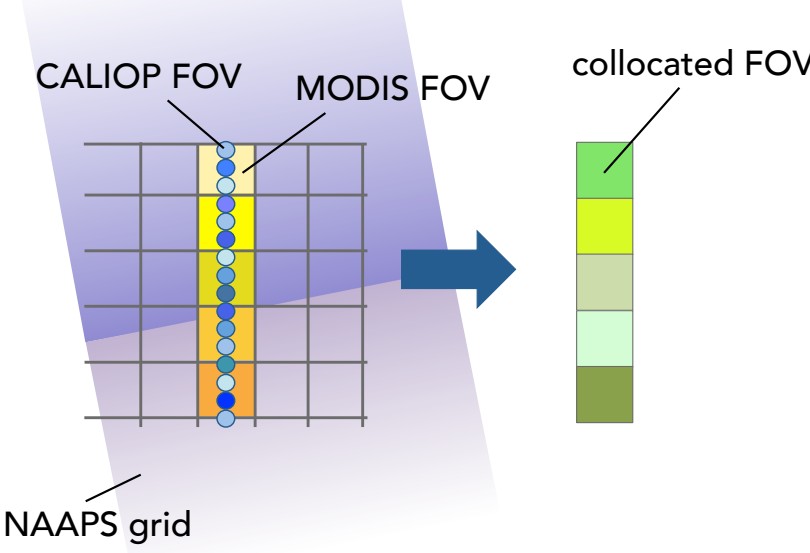

**Fig. 1.** Schematic showing the collocation process matching MODIS, CALIOP, and NAAPS. Multiple CALIOP fields of view (FOV) are mapped to a single MODIS FOV and averaged. Each satellite FOV is then assigned a NAAPS grid that is closest in space and time. This process is repeated for ten years of A-Train orbit to compose the CCARA dataset.

| mean sea level | surface elevation |
|---|---|
| 18 km | 6 km |
| 16 km | 5 km |
| 14 km | 4 km |
| ⋮ | 3 km |
| 6 km | 2.5 km |
| 5 km | 2 km |
| 4 km | 1.5 km |
| 3 km | 1 km |
| 2 km | 0.5 km |
| 1 km | 0.2 km |
| 0 km | 0 km |

**Fig. 2.** The vertical layers saved in the CCARA dataset. Left - mean sea level layers (MSL layers) are defined with respect to sea level and capture the full atmospheric column. Right - planetary boundary layer layers (PBL layers) are defined with respect to surface elevation and capture the boundary layer.

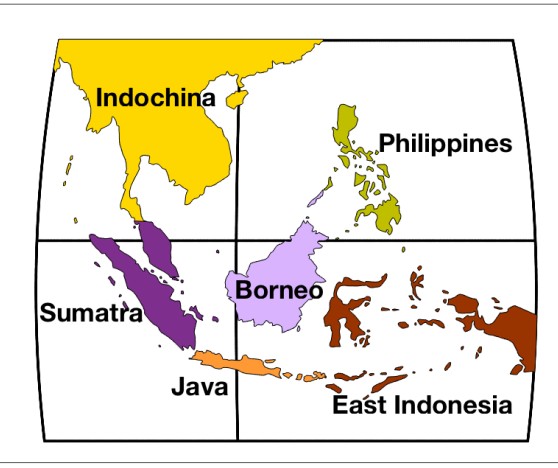
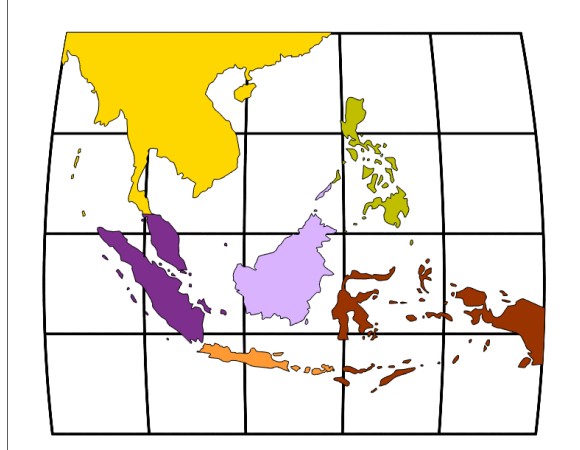

**Fig. 3.** Regions for computing AOD anomaly thresholds (left) and regions for investigating the FAIE (right). For land, the same regions are used for both AOD anomaly calculations and for the FAIE analysis. For ocean, we perform the FAIE analysis on twenty 10x10-degree grids, but aerosol anomaly thresholds for ocean regions are found using four larger grids, shown in the left plot. Land regions are distinguished with unique colors: Indochina in yellow, the Philippines in green, East Indonesia in dark red, Java in orange, Borneo in lavender, and Sumatra in purple.

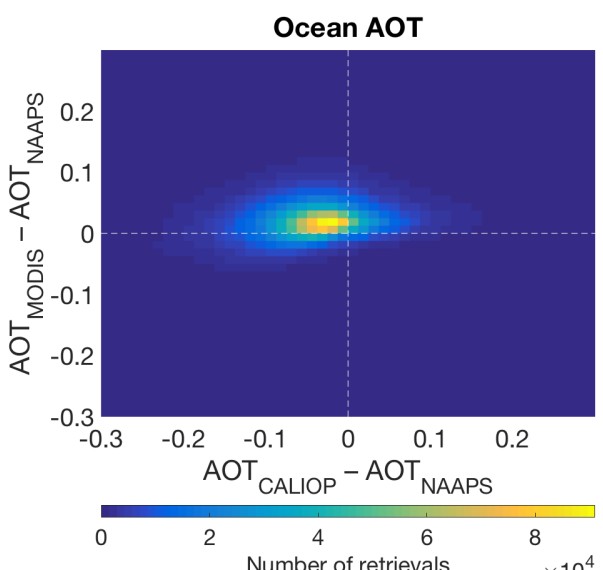
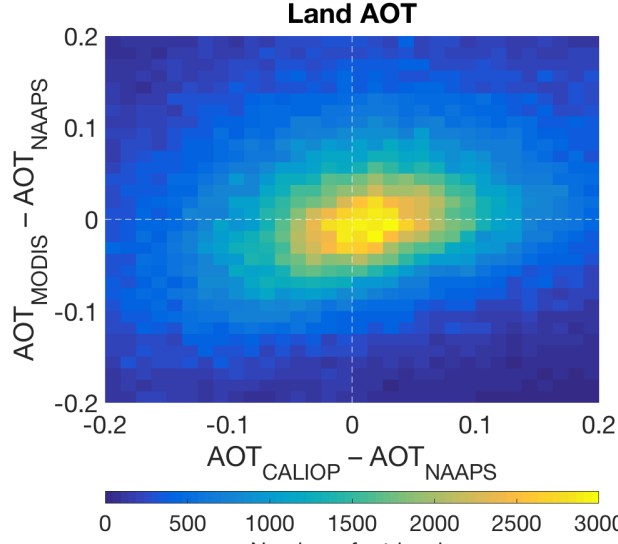

**Fig. 4.** 2D histogram of AOD innovations for ocean retrievals (left) and land retrievals (right) over SEA during 2006-2016. $AOD_{CALIOP}$ minus $AOD_{NAAPS}$ is shown on the x-axis and $AOD_{MODIS}$ minus $AOD_{NAAPS}$ is shown on the y-axis. The color scale represents the count of retrievals in each bias bin. Only cloud-cleared pixels classified by both MODIS and CALIOP are included. Only pixels where NAAPS, MODIS, and CALIOP all detect non-zero AOD are included.

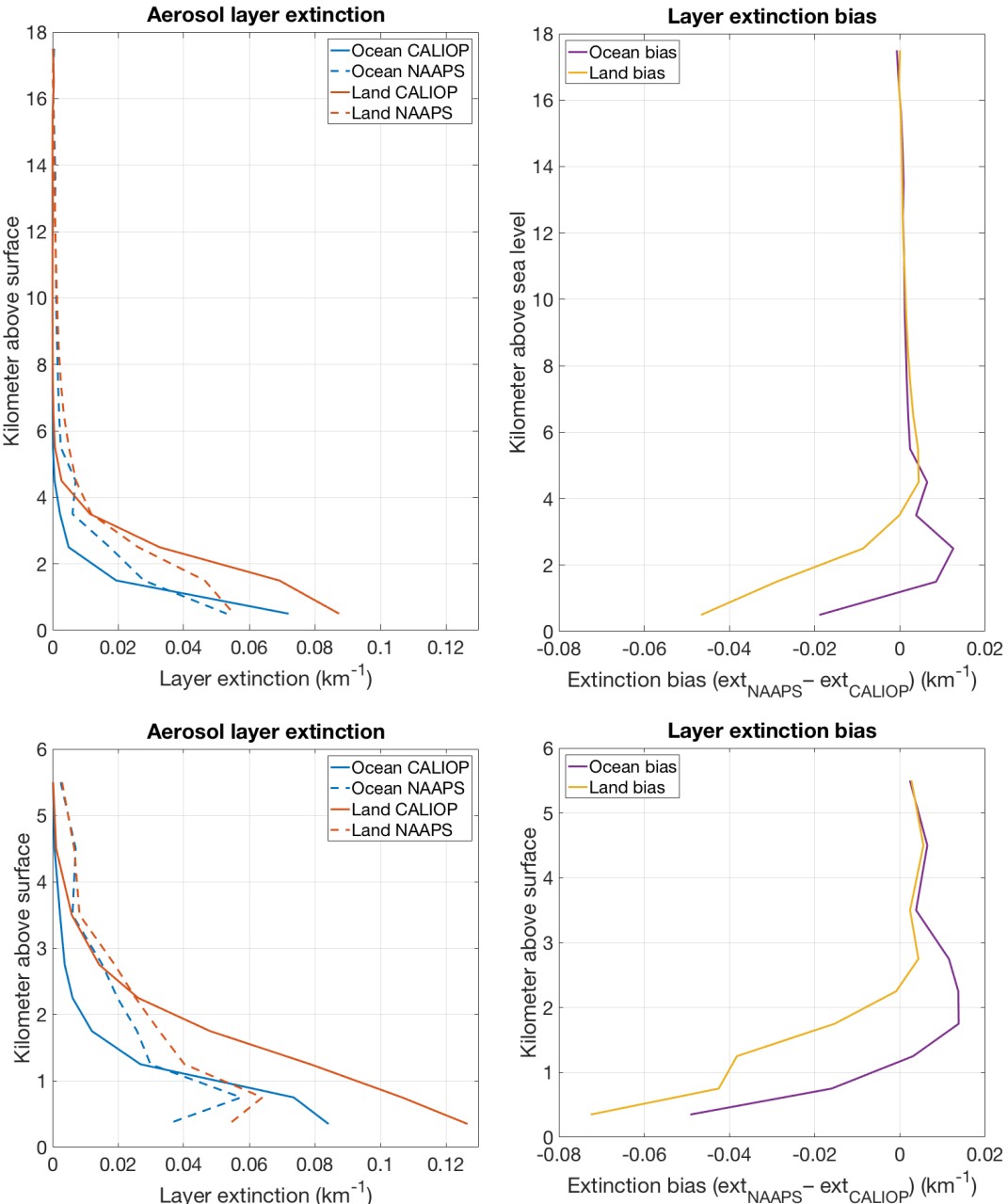

**Fig. 5.** Vertical layer distributions of average CALIOP and NAAPS aerosol extinctions (left panels) and vertical distributions of CALIOP-NAAPS extinction bias (right panels). Extinctions and extinction biases are shown for PBL layers (bottom panels) and MSL layers (top panels). Layer extinction averages are for all SEA retrievals during 2006-2016. Only CALIOP cloud-cleared pixels and pixels where NAAPS and CALIOP detect positive total AOD are included.

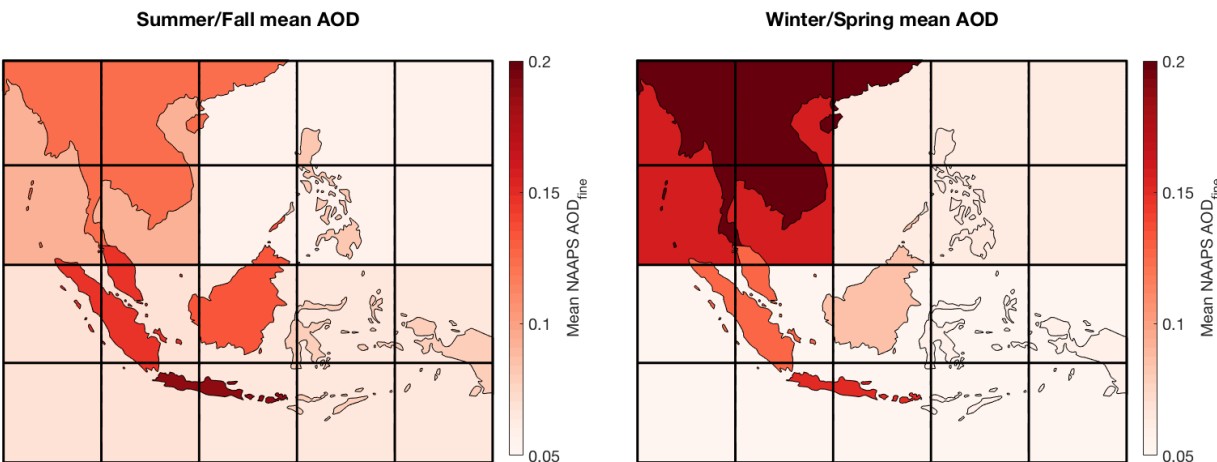

**Fig. 6.** Geometric means of fine mode AOD from NAAPS for the SEA domain separated into boreal summer and fall (JJASON) on the left and winter and spring (DJFMAM) on the right. Note the means are also separated by land and ocean.

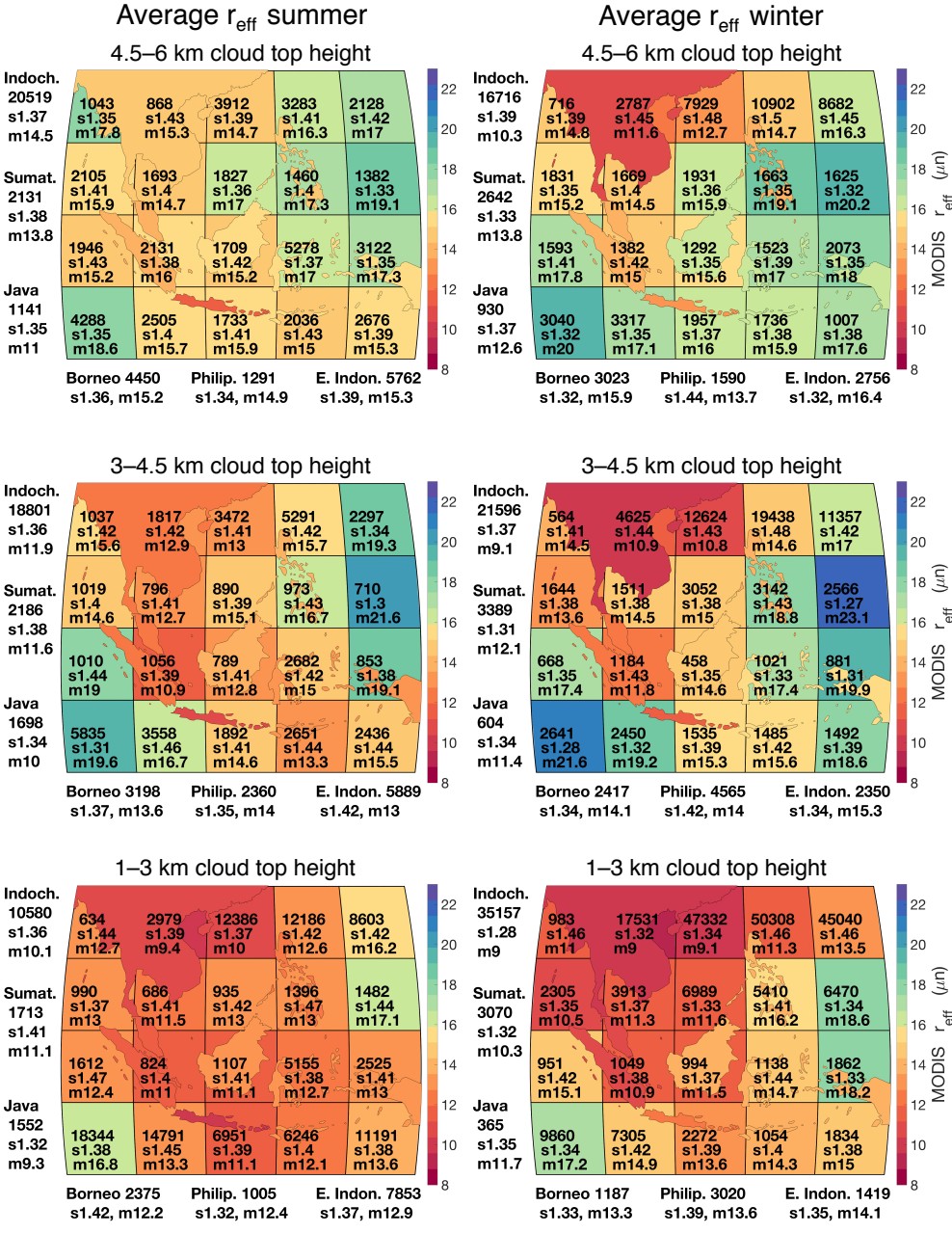

**Fig. 7:** Geometric means of baseline $r_{eff}$ for each ocean grid and land region for summer (left column) and winter (right column) spanning 2006 to 2016 for three cloud height regimes. Each region is accompanied by three black bolded numbers, the first shows the number of counts in each $r_{eff}$ population, the second is the geometric standard deviation of each population (preceded by an 's'), the third value is the median of the $r_{eff}$ population (preceded by an 'm'). Bottom panels show clouds with tops at 1-3 km in height, middle panels show clouds at 3-4.5 km, and top panels show clouds at 4.5-6 km. The colorscale is the geometric mean of the effective radius (in microns).

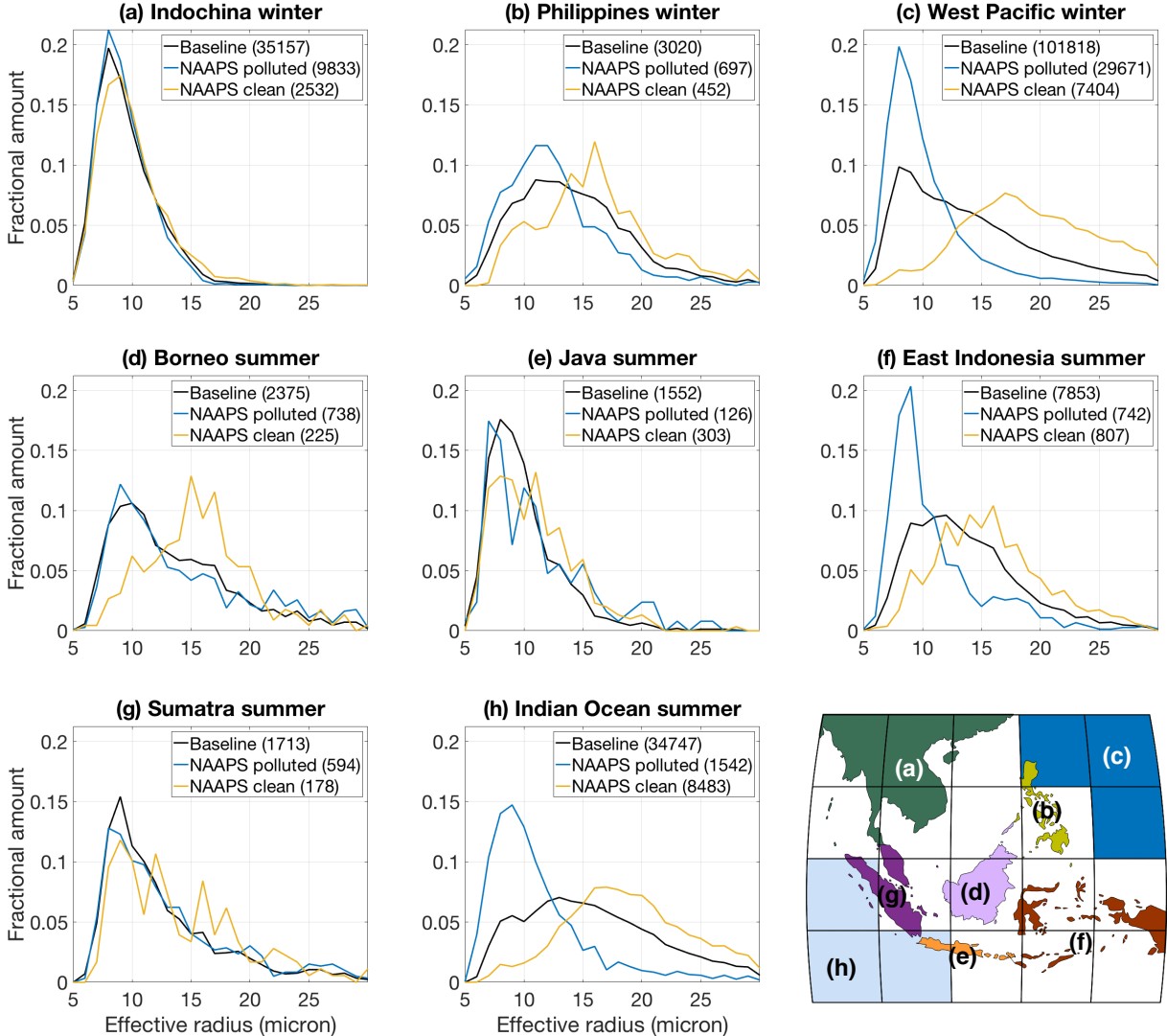

**Fig. 8.** Normalized distributions of MODIS effective radius for baseline, polluted, and clean retrievals showing six regions in panels (a) through (h): Indochina, the Philippines, and the West Pacific Ocean during winter and Borneo, Java, East Indonesia, Sumatra, and the Indian Ocean during summer. Each panel has the same x- and y-axis scale, where the y-axis shows the fractional amount of clouds. Counts of each population are shown in parentheses in the legends. A map in the lower left is provided as a reference to show each region.

## Summer (JJASON)

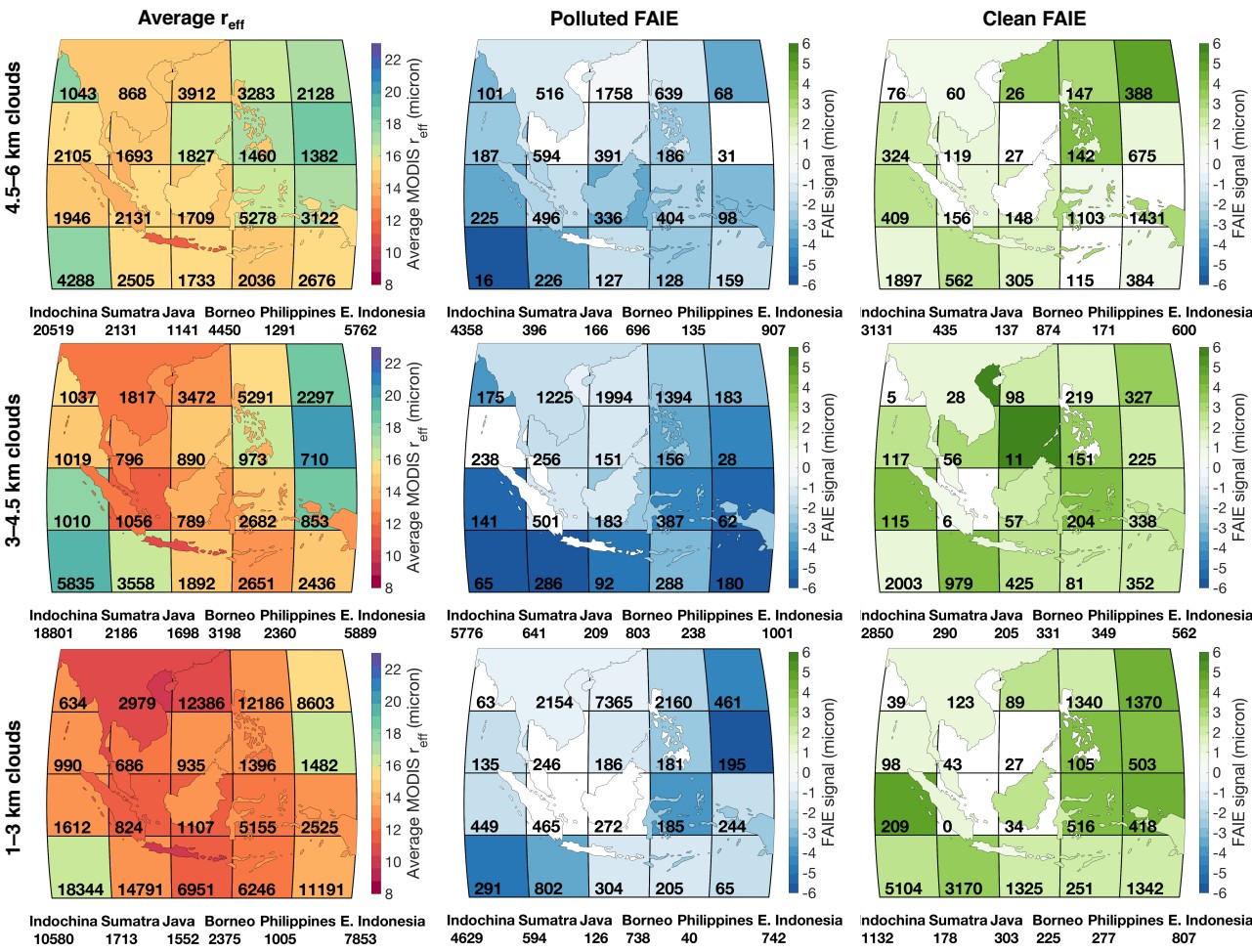

**Fig. 9.** Maps illustrating average baseline r_eff and significant polluted and clean FAIE signals. The black bolded numbers signify the count of the cloud populations. Summer months June through November are shown here. The columns from left to right show the average bulk (or baseline) r_eff, significant polluted FAIE, and significant clean FAIE. The rows from bottom to top show cloud top height regimes: clouds between 1-3 km, 3-4.5 km, and 4.5-6 km. Polluted and clean FAIE signals are shaded only if the significance t-test between baseline and anomalous r_eff populations yields a p-value < 0.01.

## Winter (DJFMAM)

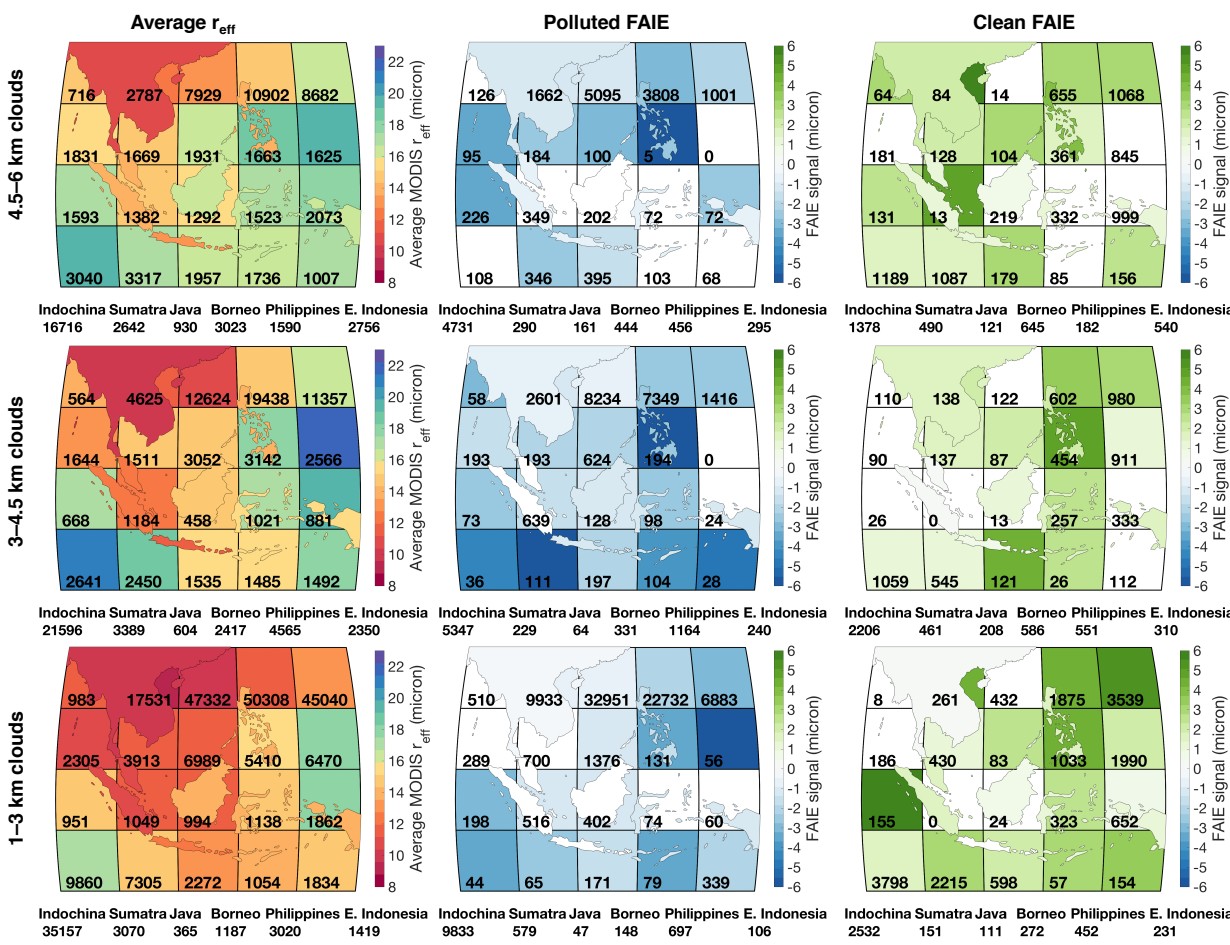

**Fig. 10.** Identical to Fig. 9 but for winter months December through May.