# Peer review of "Exploring the First Aerosol Indirect Effect over Southeast Asia Using a Ten-Year Collocated MODIS, CALIOP, and Model Dataset"

_Atmospheric Chemistry and Physics, 2018_

## Referee Comment (RC1) · Anonymous Referee #1 · 16 Apr 2018

This manuscript combined satellite observations and model reanalysis data (NAAPS) to study the relationship between aerosol and cloud effective radius (re) over Southeast Asia. They used both MODIS, CALIPSO, and model results to present the collocated dataset for analysis of aerosol-cloud relationship, which is critical for this kind of scientific topic. However, there have some fundamental issues need to be addresses carefully. First, they used aerosol information from the model simulations, since MODIS retrieval only provides few retrievals in moderate and high cloudy environment, and CALIPSO is unable to see aerosol below a cloud layer under some conditions, as the authors stated. The model can indeed provide much more aerosol information, including spatial-temporal distribution, size distribution, chemical component, etc., but the

method to collocate aerosol from model and cloud from satellite retrieval are unclear, especially the model results are 6 hourly products, and how to collocate with the time of satellite overpass? the model spatial resolution is much lower than satellite retrieval, how to collocate them in spatial? This need to be clarify. Second, it is unclear that how accurate is the modeled aerosol compared to the satellite retrieval. The authors gave the statistical analysis of the AOD differences between the model and satellite, as well as the vertical profile, but what about the spatial distribution of the model results in comparison to satellite retrievals? Also, only the relationship between aerosol and cloud effective radius is studied, and definition of first aerosol indirect effect (FIRE) on p.15 and use of this definition in the title is inappropriate. The results and conclusions are not quite new. The anti-correlation of aerosol and cloud effective radius over the oceanic regions have been reported extensively.

Specific comments: p.3, bottom line, what aerosol amount? Please clarify.

p.4, line 12-15, the study of Ma et al. (2010) was cited in the text, but not listed in reference.

Section 2.3, p. 6, line 9 15, it is unclear that how to collocate the data, e.g. line 19-10, level 1 or level 2, or both? Also, on the collocation of time. As described in sec. 2.2 (p.5), the model results are 6-hourly product, so how to collocate the model results with the time of satellite overpasses?

p.8, line line 5-9, what is the average of AOD over the region?

Figure. 5, top panel is for above sea level, not surface.

p.12, line 1-2, the definition of aerosol anomaly is difficult to understand. 'when the AOD is greater than an upper threshold, ($\mu$AOD $\times$ $\sigma$AOD). A negative aerosol anomaly, or clean event, is when the AOD is less than a lower threshold, ($\mu$AOD Ãů $\sigma$AOD)', why the polluted and clean condition are defined like this?

Figure. 7: . . .in the lower left", should be lower right.

p.15, top of the page, the definition of FAIE. This definition is only the differences of cloud effective radius between the polluted (clean) conditions and baseline. This is actually not FAIE, so the definition is incorrect and misleading.

p. 17, line 7: aerosol amount, what aerosol properties is used in the analysis, AOD or aerosol number? or something else?

p.17, line 10-11, finding. . ., this is not new.

---

## Referee Comment (RC2) · Anonymous Referee #3 · 16 Apr 2018

Ross et al. (2018) uses ten years of satellite-based observations and NAAPS reanalysis data to examine the first aerosol direct effect in Southeast Asia (SEA). The combined satellite-based dataset (termed "CCARA" or "Curtain Cloud-Aerosol Regional A-Train") is an invaluable resource for aerosol-cloud research over Southeast Asia (SEA), given the care taken to screen cloud contamination in the cloud-endemic region and assemble the observations into a coherent package. I am looking forward to see more future work based on CCARA. However, the following general and specific comments should be addressed for this manuscript. Suggested changes are also included in uploaded pdf.

[Figure]

General comments:

1) Comprehensive literature review on related studies in SEA (besides classics on aerosol effects and authors' publications) is needed to put the work in context. E.g., Lin et al. (2014) gave an overview on interactions between biomass burning aerosols and clouds over SEA, and Lee et al. (2014) modelled the impact of aerosols on atmospheric circulation and rainfall over SEA.

2) I think the structure of the paper is slightly skewed at the moment. The paper devotes almost half of the pages to introduction and methods, and the other half to results. Given that the paper describes a new dataset, a comprehensive coverage to the methods is quite understandable. That being said, the discussions on the results are thus comparatively limited. I think more discussions of these results in relation to the biomass burning and pollution situation in SEA are possible. Pls see specific comments for more details.

3) I think the authors can make better use of the figures to expand the discussions. E.g., panels in the figures can be labelled "a, b, c...". Firstly, this will help the readers to relate the in-text discussions to the different panels in the figures or features within the panels. Secondly, the authors may also find it easier to focus on specific panels to highlight point of interests. Pls see specific comments for more details.

E.g., the punchline of the paper in my opinion is Figure 8 and 9 and are discussed in only two pages out of 17 pages of text, focusing mainly on the statistics on what made the FAIE signal significant. There are maybe two paragraphs (less than a page) on how these results relate to biomass burning and pollution over the region. Moreover, these two figures are rather complicated for the uninitiated, and consists spatial, temporal and vertical info for 6 land regions and 20 sea regions (4 aggregated sea regions). The authors should expand the discussions to take the readers through the richness of these two figures.

Are there opportunities to explore how the variations of injection heights of smoke and pollution impact the cloud reff at each level? E.g. look carefully at Sumatra and Java for Figure 9. reff for Sumatra gets warmer (decreases) downwards vertically, but reff for Java seems be warmest (lowest reff) at 3 - 4.5 km. Perhaps the authors can explore if this be due to convective pumping bringing the pollution (not smoke) to that level? Do note that these are suggestions to take the discussion further in more details.

4) The authors should refer to ACP manuscript preparation guidelines for authors (at https://www.atmospheric-chemistry-and-physics.net/for_authors/manuscript_preparation.html) when preparing manuscript.

E.g. Pg 6 Ln 25: "as illustrated in Figure 1" should be "as illustrated in Fig. 1" according to guidelines.

Another e.g. within the same line: "333-m" should be "333 m" according to house standard not to hyphenate modifiers containing abbreviated units.

The authors should also check the language within the manuscript (some suggestions are given in the uploaded pdf). There are also numerous use of semicolons which break up the flow in the text (pls see uploaded pdf as well).

Specific comments:

Title: The entire SEA is effectively covered in the study. It will be appropriate to title the manuscript as such.

Abstract: You may want to introduce CCARA here to the scientific community. This is important as interested readers will pick it up immediately and possibly relate to future work using the dataset. You don't want CCARA to be lost in the rest of the text.

Pg 2 Ln 17: I have not seen this term "Greater Southeast Asia" being used in literature. Pls remove it.

Pg 6 Ln 1 - 3: I understand that ECMWF met data performs well for SEA, but it is equally important that NAVGEM performs well too as it is used to drive aerosol transport used in the study. The authors should provide evidence that the mismatch between met and modelled aerosol parameters is acceptable (e.g., statistics from the data) or only the mentioned key parameters of e.g., wind shear, water vapor, are important for the phenomenon being studied (e.g., citations).

Section 3.1.1: Injection heights of biomass burning smoke is extremely important for determining the vertical distribution of biomass burning aerosols. How is the injection height in NAAPS determined? Does the method matter for the different varieties of "biomass burning" present in SEA (Reid et al., 2013)?

For SEA, biomass burning emissions does not seemed to be directly injected into the middle to upper troposphere (Campbell et al., 2013; Tosca et al., 2011). As mentioned by Reid et al. (2013), the type of "biomass burning" in SEA actually varies a lot, thus possibly resulting in a wide range of injection heights. The authors should discuss the bias for CALIOP-NAAPS in more details framed in the context of the above-mentioned papers. The model results seems to be higher than the CALIPSO results, especially for ocean (Fig. 5).

Pg 8 Ln 9 - 16: The four larger ocean grids (e.g., ocean - south west) mentioned here should be named. They are later mentioned in Table 1 without reference back to Figure 3. A single sentence in Section 3.1.2 like e.g., "the north west, north east, south west and south east of ocean mentioned in Table 1 are with reference to the larger ocean grids (Fig. 3a)." will help the readers much in understanding the analysis.

Pg 11 Ln 4 and 9: Campbell et al. (2012) are cited in text while referenced as Campbell et al. (2013) in reference seciton. Pls kindly check through text for citation errors.

Section 3.1.3: Where is West Pacific (Pg 13 Ln 9)? This is an example of figures that could be used more effectively. I suppose the region to look at is the 3 grids at the top right hand corner of each panels (similar to West Pacific as mentioned in Figure 7 bottom panel). If the boundaries of the 3 grids can be made bold, it will make it easier for the readers to follow. Pls identify similar issues with the use of figures within the manuscript and make changes. I have highlighted a few (pls see uploaded pdf).

Section 3.2.2: It is difficult for the readers to follow the impact of smoke on reff without visualizing the spatial distributions of smoke transport (or plumes) over SEA. It may be good to include NAAPS AOD plots for boreal summer and winter in the manuscript as well to orientate the readers to the direction of smoke transport.

Figure 6: Pls consider swapping info for "E. Indon." with "Philip." When the reader go through the info, they would expect it to be presented in a logical manner. If you swap info for "E. Indon." with "Philip.", you will be going through the land regions in a counter clockwise way, with Indochina at the top, moving south to Sumatra, moving east to Java, followed by Borneo and E. Indonesia and finally, moving north to Philippines. Currently, it seems you are moving counter clockwise, then suddenly zipping to Philippines on top, before going back to E. Indonesia.

Figure 7: These panels are of great interest to the region. Although it is not possible to present everything within the manuscript, but the summer and winter plots for every land and sea region should be included in a supplement.

Figure 8 and 9: The authors mentioned that the FAIE signal maps are shaded if the FAIE signal is deemed statistically significant. The problem is when you allocate white (or a pale color) to 0 FAIE signal in the color bar, it is difficult to differentiate whether an area is shaded or not.

**Supplement:**

[revised manuscript text omitted]

We define a positive aerosol anomaly, or polluted event, when the AOD is greater than an upper threshold, ($\mu_{AOD} \times \sigma_{AOD}$). A negative aerosol anomaly, or clean event, is when the AOD is less than a lower threshold, ($\mu_{AOD} \div \sigma_{AOD}$). Winter and summer upper and lower AOD thresholds for each region are shown in Table 1, along with the median AOD values. Recall that to define ocean aerosol anomalies, the region is split into four large grids, which we will distinguish by referring to as southwest, northwest, southeast, and northeast grids. Note the seasonal differences in each region and recall that Indochina experiences a  season during winter months while Java, Sumatra, and Borneo experience a  season during summer months. These burning seasons are reflected in our AOD thresholds. Summer upper AOD thresholds for Java, Borneo, and Sumatra are much greater than their winter upper thresholds. Over Indochina, the winter upper AOD threshold over double that in the summer.

| | Boreal Summer | | | Boreal Winter | | |
|---|---|---|---|---|---|---|
| | $\mu_{AOD} \div \sigma_{AOD}$, median, $\mu_{AOD} \times \sigma_{AOD}$ | | | $\mu_{AOD} \div \sigma_{AOD}$, median, $\
[revised manuscript text omitted]

Moving to the boreal summer, the monsoon flips to more wet conditions to the north and drier more fire prone conditions to the south. In Panel D, Borneo behaves as would be expected owing to its relatively high population, little sensitivity in the baseline and more polluted conditions, but with some residual sensitivity for the "cleanest" of conditions. Such cleaner conditions do not  on the highly populated island of Java, with considerably less sensitivity to modeled perturbations in AOD. Similar findings were found for Sumatra (g). Some of the largest spreads in $r_{eff}$ for boreal summer were found in the more remote areas, such as Eastern Indonesia over land (f) or the Western Indian ocean – a frequent receptor for pollution from Java (h).

While the examples in Figure 7 hint at signals of clean and polluted FAIE signals, we must consider the statistical significance of these distributions in the comparisons. The next section will systematically explore clean and polluted FAIE signals for each region in SEA and investigate signals of the FAIE where statistical testing has deemed the FAIE significant.

**3.2.2 Mapped FAIE for winter and summer**

In this section, we compute geometric means of $r_{eff}$ for baseline, polluted, and clean retrievals for each region, season, and cloud height regime. We use a t-test to determine whether the geometric means of those distributions are significantly different from one another to yield a FAIE signal. The significance test is designed such that we can test specifically if the clean $r_{eff}$ is *larger* than the baseline $r_{eff}$ (a clean FAIE) and if the polluted $r_{eff}$ is *smaller* than the baseline $r_{eff}$ (polluted FAIE). The magnitude of the FAIE is the displacement of the anomalous (clean or polluted) $r_{eff}$ from the baseline $r_{eff}$:

[Figure]

[Figure]

$$\text{clean FAIE} = \text{clean } r_{eff} - \text{baseline } r_{eff} \geq 0$$

$$\text{polluted FAIE} = \text{polluted } r_{eff} - \text{baseline } r_{eff} \leq 0$$

This definition is defined so the clean FAIE is positive in sign and the polluted FAIE is negative. Figure 8 shows maps of average $r_{eff}$ values for baseline populations, as well as maps of polluted and clean FAIE signals during summer months.

Winter months are shown in Figure 9. The rows in this figure signify different cloud top height regimes; the columns show the baseline $r_{eff}$, polluted FAIE signal, and clean FAIE signal. The cloud height regimes signified by the rows in Figures 8 and 9 correlate from bottom to top: 1–3 km, 3–4.5 km, and 4.5–6 km. The FAIE signal maps are only shaded if the FAIE is deemed statistically significant ($p < 0.01$). Note that the same region may yield a significant clean FAIE but not a significant polluted FAIE, or vice versa.

During winter months shown in Figure 9, for clouds nearest to the surface (between 1–3 km) there is a significant clean and polluted FAIE (up to 6 microns in magnitude) for the West Pacific Ocean retrievals, just east of the Philippines. This suggests that under heavily polluted environments, the average effective radius is up to 6 microns smaller relative to the effective radius under every-day scenes. Under extremely clean scenes, the average effective radius can be over 5 microns larger than the bulk average. Even for high clouds at 4.5–6 km, a polluted and clean FAIE signal is found for some of the

West Pacific grids. A polluted and clean FAIE is also present for the Eastern Indian Ocean. A very weak (< 1 micron in magnitude) clean and polluted FAIE signal exists for low clouds over Indochina. These FAIE signals are weak as the environment is likely almost always saturated with fine mode aerosol, resulting in additional sources having limited impact on the water cloud droplet size. Comparing Indochina in Figure 8 to Panel (a) in Figure 7, note that while the normalized histograms of clean and polluted $r_{eff}$ are almost identical, the slight difference in the right-side tails of this distribution, in addition to the high number of counts, yields a p-value small enough to consider this difference significant. The example of wintertime Indochina is consistent with the notion that there is a physical upper limit to cloud-aerosol interactions (Painemal and Zuidema, 2013). A slightly stronger FAIE over Indochina does appear for higher altitude clouds, though the average $r_{eff}$ for all (baseline) clouds over Indochina stays quite small and does not exceed 12 microns even for the tallest clouds.

The black bolded numbers in Figures 8 and 9 signify the count of the cloud populations for each region in each panel. The purpose of showing these counts is to indicate whether the lack of a FAIE signal is due to low cloud counts or an insignificant change in the average anomalous $r_{eff}$.

Over some West Pacific peripheral ocean regions in Figure 9, there is a lack of polluted liquid $r_{eff}$ retrievals for higher clouds (3–6 km). One reason for this is that the region is regularly covered in cirrus, reducing the likelihood of a liquid retrieval from MODIS. Additionally, while significant smoke and pollution can reach this region, it is somewhat rare and compounded by how we defined aerosol anomaly. Our aerosol thresholds were defined by determining whether a certain AOD amount was anomalous relative to all clouds in that region during that season, *not* answering whether the AOD was anomalous to specific cloud top heights in that region during that season. Further, recall that ocean aerosol anomalies are determined for larger ocean grids, not the small 10x10-deg grids in Figures 8 and 9. For these reasons, it follows that some

[revised manuscript text omitted]

---

## Author Comment (AC1) · 9 Jul 2018

**Author responses to comments on "Exploring the First Aerosol Indirect Effect over the Maritime Continent Using a Ten-Year Collocated MODIS, CALIOP, and Model Dataset" by Ross et al.**

**Anonymous Referee #1 General Comments**

This manuscript combined satellite observations and model reanalysis data (NAAPS) to study the relationship between aerosol and cloud effective radius (re) over Southeast Asia. They used both MODIS, CALIPSO, and model results to present the collocated dataset for analysis of aerosol-cloud relationship, which is critical for this kind of scientific topic. However, there have some fundamental issues need to be addresses carefully. First, they used aerosol information from the model simulations, since MODIS retrieval only provides few retrievals in moderate and high cloudy environment, and CALIPSO is unable to see aerosol below a cloud layer under some conditions, as the authors stated. The model can indeed provide much more aerosol information, including spatial-temporal distribution, size distribution, chemical component, etc., but the method to collocate aerosol from model and cloud from satellite retrieval are unclear, especially the model results are 6 hourly products, and how to collocate with the time of satellite overpass? the model spatial resolution is much lower than satellite retrieval, how to collocate them in spatial? This need to be clarify. Second, it is unclear that how accurate is the modeled aerosol compared to the satellite retrieval. The authors gave the statistical analysis of the AOD differences between the model and satellite, as well as the vertical profile, but what about the spatial distribution of the model results in comparison to satellite retrievals? Also, only the relationship between aerosol and cloud effective radius is studied, and definition of first aerosol indirect effect (FIRE) on p.15 and use of this definition in the title is inappropriate. The results and conclusions are not quite new. The anti-correlation of aerosol and cloud effective radius over the oceanic regions have been reported extensively.

**Response to Anonymous Referee #1 General Comments**

The referee comments expressed concern over the method of collocation between satellite retrievals and model output. The authors address the method of collocation in a simplistic manner: after a MODIS-CALIOP FOV is collocated, we find which 6-hourly 1x1-degree grid fits 'closest' in space and time. The closest model \1x1-deg grid is then assigned to the FOV. This is illustrated in Figure 1. The referee's comments also addressed uncertainty within the model aerosol and the satellite observed aerosol. As stated in the manuscript., quality-controlled MODIS aerosol retrievals are assimilated into the NAAPS reanalysis. However, due to the near ubiquitous cloud cover in SEA, MODIS aerosol retrievals in the region are infrequent. Therefore, we provide AOD bias statistics between the observations and model to baseline relative model performance which correlates closely with MODIS.

The referee questions if a higher resolution model would provide a better information to compare to the satellite retrievals. However, over a large domain such as SE Asia the exact opposite is true. There are numerous perturbations to the aerosol environment from a host of multi-scale processes, that cannot be reproduced in any model, nor even monitored from any available system. Rather, here we take more of a regime approach. When the model suggests above or below average particle concentrations, over all we find decreased and increased effective radius, respectively. In regions with low average concentrations, the effect is stronger than regions with high average concentrations. Through such analyses, we can subsequently project regionally into climate change timescales. This leads to the second major comment on the *First Aerosol Indirect Effect.*

We understand that the use of the term *First Aerosol Indirect Effect* is not precisely correct, since manuscript does not explore the radiative aerosol indirect effect but instead focuses on the cloud-aerosol interactions. However, the anti-correlation of aerosol amount and droplet size (effective radius) is the exact mechanism through which the

*FAIE* is theorized. We also argue that the results and conclusions are quite new. The paper shows that throughout Southeast Asia, effects of aerosols on maritime clouds have far reaching effects. Additionally, this paper shows that over land areas during seasons of biomass burning (such as Indochina during winter months), distributions of relatively clean and relatively polluted cloud effective radii are extremely similar with the hyposthis being that regions are already saturated with aerosol even the for relatively "clean" time periods. This is in contrast to maritime regions (i.e. West Pacific winter), where relatively clean and relatively polluted clouds have very different effective radius distributions (Figure 7 panels a and c). The conclusions of this paper go beyond a simple anti-correlation between aerosol and effective radius, they provide detailed insight into cloud-aerosol interactions in an area of the world that is very hard to both observe and model by combining model and observations in a unique way.

**Anonymous Referee #1 Specific Comments**

p.3, bottom line, what aerosol amount? Please clarify.

p.4, line 12-15, the study of Ma et al. (2010) was cited in the text, but not listed in reference.

Section 2.3, p. 6, line 9 15, it is unclear that how to collocate the data, e.g. line 19-10, level 1 or level 2,
or both? Also, on the collocation of time. As described in sec. 2.2 (p.5), the model results are 6-hourly product, so how to collocate the model results with the time of satellite overpasses?

p.8, line line 5-9, what is the average of AOD over the region?

Figure. 5, top panel is for above sea level, not surface.

p.12, line 1-2, the definition of aerosol anomaly is difficult to understand. 'when the AOD is greater
than an upper threshold, ($\mu AOD \times \sigma AOD$). A negative aerosol anomaly, or clean event, is when the AOD is less than a lower threshold, ($\mu AOD/\sigma AOD$)', why the polluted and clean condition are defined like this?

Figure. 7: . . .in the lower left", should be lower right.

p.15, top of the page, the definition of FAIE. This definition is only the differences of cloud effective
radius between the polluted (clean) conditions and baseline. This is actually not FAIE, so the definition is incorrect and misleading.

p. 17, line 7: aerosol amount, what aerosol properties is used in the analysis, AOD or aerosol number? or something else?

p.17, line 10-11, finding. . ., this is not new.

**Responses to Anonymous Referee #1 Specific Comments**

*p.3, bottom line, what aerosol amount? Please clarify.*
As mentioned in the manuscript, the term 'aerosol amount' is used to infer NAAPS AOD. However, since NAAPS
AOD is determined using aerosol mass concentrations, assimilated with MODIS and MISR AOD, the term aerosol
amount was appropriate.

*p.4, line 12-15, the study of Ma et al. (2010) was cited in the text, but not listed in reference.*
Thank you for pointing this out, I will add Ma et al. to the references.

*Section 2.3, p. 6, line 9 15, it is unclear that how to collocate the data, e.g. line 19-10, level 1 or level 2, or both? Also, on the collocation of time. As described in sec. 2.2 (p.5), the model results are 6-hourly product, so how to collocate the model results with the time of satellite overpasses?*
These concerns were addressed in the response to general comments above. Section 2.3.1 further explains the
collocation process between satellite observations and model output. Regarding the matching in time, once the
collocation between MODIS and CALIOP is performed, that FOV has an assigned overpass time (MODIS
timestamp). We simply find which 6-hourly model field is closest in time to the MODIS timestamp. I will add this
clarification into section 2.3.1.

*p.8, line line 5-9, what is the average of AOD over the region?*
We are unsure what your comment is referring to, but I believe it is p. 9 in section 2.4. I will refer you to table 3.1
where we list median of AOD for each region and season in this paper.

*Figure. 5, top panel is for above sea level, not surface.* Thank you for pointing out this typo,
I will correct the y-axis text in Figure 5.

*p.12, line 1-2, the definition of aerosol anomaly is difficult to understand. 'when the AOD is greater than an upper threshold, ($\mu AOD \times \sigma AOD$). A negative aerosol anomaly, or clean event, is when the AOD is less than a lower threshold, ($\mu AOD / \sigma AOD$)', why the polluted and clean condition are defined like this?* The aerosol anomalies are
defined as such because what is anomalous for one region/season isn't necessarily anomalous for another. Upper
and lower thresholds are defined for each region/season because what is considered 'clean' in a polluted land region
like Indochina should not also be considered 'clean' in a maritime region like the West Pacific. Average AOD and
therefore aerosol anomalies must be defined individually for each region/season.

*Figure. 7: . . .in the lower left", should be lower right.*
Thank you for pointing this out, I will change the caption text in Figure 7.

*p.15, top of the page, the definition of FAIE. This definition is only the differences of cloud effective radius between the polluted (clean) conditions and baseline. This is actually not FAIE, so the definition is incorrect and misleading.*
You are correct in pointing out that the full definition of the FAIE includes an albedo effect. In this paper, we only
discuss the mechanism which drives the FAIE, namely the anti-correlation between effective radius and AOD.
These concerns are addressed in the general comments above. As the cloud effective radius has a direct effect on
the albedo we will therefore leave the FAIE terminology in the paper.

*p. 17, line 7: aerosol amount, what aerosol properties is used in the analysis, AOD or aerosol number? or something else?*
As mentioned in Section 2.4, aerosol amount is determined by NAAPS AOD.

*p.17, line 10-11, finding..., this is not new.*
We hold that aerosol effects from land regions in SEA do have far reaching effects on marine clouds. We believe this result to be original and new. We have addressed these concerns in the response to the general comments.

**Anonymous Referee #3 General Comments**

1) Comprehensive literature review on related studies in SEA (besides classics on aerosol effects and authors' publications) is needed to put the work in context. E.g., Lin et al. (2014) gave an overview on interactions between biomass burning aerosols and clouds over SEA, and Lee et al. (2014) modelled the impact of aerosols on atmospheric circulation and rainfall over SEA.

2) I think the structure of the paper is slightly skewed at the moment. The paper devotes almost half of the pages to introduction and methods, and the other half to results. Given that the paper describes a new dataset, a comprehensive coverage to the methods is quite understandable. That being said, the discussions on the results are thus comparatively limited. I think more discussions of these results in relation to the biomass burning and pollution
situation in SEA are possible. Pls see specific comments for more details.

3) I think the authors can make better use of the figures to expand the discussions. E.g., panels in the figures can be labelled "a, b, c...". Firstly, this will help the readers to relate the in-text discussions to the different panels in the figures or features within the panels. Secondly, the authors may also find it easier to focus on specific panels to
highlight point of interests. Pls see specific comments for more details. E.g., the punchline of the paper in my opinion is Figure 8 and 9 and are discussed in only two pages out of 17 pages of text, focusing mainly on the statistics on what made the FAIE signal significant. There are maybe two paragraphs (less than a page) on how these results relate to biomass burning and pollution over the region. Moreover, these two figures are rather complicated for the uninitiated, and consists spatial, temporal and vertical info for 6 land regions and 20 sea regions
(4 aggregated sea regions). The authors should expand the discussions to take the readers through the richness of these two figures. Are there opportunities to explore how the variations of injection heights of smoke and pollution impact the cloud reff at each level? E.g. look carefully at Sumatra and Java for Figure 9. reff for Sumatra gets warmer (decreases) downwards vertically, but reff for Java seems be warmest (lowest reff) at 3 - 4.5 km. Perhaps the authors can explore if this be due to convective pumping bringing the pollution (not smoke) to that level? Do
note that these are suggestions to take the discussion further in more details.

4) The authors should refer to ACP manuscript preparation guidelines for authors (at https://www.atmospheric-chemistry-and- physics.net/for_authors/manuscript_preparation.html) when preparing manuscript. E.g. Pg 6 Ln 25: "as illustrated in Figure 1" should be "as illustrated in Fig. 1" according to guidelines. Another e.g. within the same
line: "333-m" should be "333 m" according to house standard not to hyphenate modifiers containing abbreviated units.
The authors should also check the language within the manuscript (some suggestions are given in the uploaded pdf. There are also numerous uses of semicolons which break up the flow in the text (pls see uploaded pdf as well).

## Response to Anonymous Referee #3 General Comments

We appreciate your thoughtful comments on our manuscript. We have incorporated some of your citation recommendations. The significance of the results were slightly expanded upon in Sections 3 and 4 of the paper, though we felt we got most of our points across concerning the anti-correlation of $r_{eff}$ and AOD and how the anti- correlation related to seasons of biomass burning. The updated manuscript includes more discussion of the key figures (now Figs. 8, 9, and 10). We have also added a mean AOD figure for each season/reason, as per your suggestion (Fig. 6). Thank you for the corrections on the ACP manuscript preparation guidelines. We made the necessary corrections. We have also adopted most of your suggestions in the supplemental draft.

Anonymous Referee #3 Specific Comments
Title: The entire SEA is effectively covered in the study. It will be appropriate to title the manuscript as such.

Abstract: You may want to introduce CCARA here to the scientific community. This is important as interested readers will pick it up immediately and possibly relate to future work using the dataset. You don't want CCARA to be lost in the rest of the text.

Pg 2 Ln 17: I have not seen this term "Greater Southeast Asia" being used in literature. Pls remove it.

Pg 6 Ln 1 - 3: I understand that ECMWF met data performs well for SEA, but it is equally important that NAVGEM performs well too as it is used to drive aerosol transport used in the study. The authors should provide evidence that the mismatch between met and modelled aerosol parameters is acceptable (e.g., statistics from the data) or only the mentioned key parameters of e.g., wind shear, water vapor, are important for the phenomenon being studied (e.g., citations).

Section 3.1.1: Injection heights of biomass burning smoke is extremely important for determining the vertical distribution of biomass burning aerosols. How is the injection height in NAAPS determined? Does the method matter for the different varieties of "biomass burning" present in SEA (Reid et al., 2013)?

For SEA, biomass burning emissions does not seemed to be directly injected into the middle to upper troposphere (Campbell et al., 2013; Tosca et al., 2011). As mentioned by Reid et al. (2013), the type of "biomass burning" in SEA actually varies a lot, thus possibly resulting in a wide range of injection heights. The authors should discuss the bias for CALIOP-NAAPS in more details framed in the context of the above-mentioned papers. The model results seems to be higher than the CALIPSO results, especially for ocean (Fig. 5).

Pg 8 Ln 9 - 16: The four larger ocean grids (e.g., ocean - south west) mentioned here should be named. They are later mentioned in Table 1 without reference back to Figure 3. A single sentence in Section 3.1.2 like e.g., "the north west, north east, south west and south east of ocean mentioned in Table 1 are with reference to the larger ocean grids (Fig. 3a)." will help the readers much in understanding the analysis.

Pg 11 Ln 4 and 9: Campbell et al. (2012) are cited in text while referenced as Campbell et al. (2013) in reference seciton. Pls kindly check through text for citation errors.

Section 3.1.3: Where is West Pacific (Pg 13 Ln 9)? This is an example of figures that could be used more effectively. I suppose the region to look at is the 3 grids at the top right hand corner of each panels (similar to West Pacific as mentioned in Figure 7 bottom panel). If the boundaries of the 3 grids can be made bold, it will make it easier for the readers to follow. Pls identify similar issues with the use of figures within the manuscript and make changes. I have highlighted a few (pls see uploaded pdf).

Section 3.2.2: It is difficult for the readers to follow the impact of smoke on reff without visualizing the spatial distributions of smoke transport (or plumes) over SEA. It may be good to include NAAPS AOD plots for boreal summer and winter in the manuscript as well to orientate the readers to the direction of smoke transport.

Figure 6: Pls consider swapping info for "E. Indon." with "Philip." When the reader go through the info, they would expect it to be presented in a logical manner. If you swap info for "E. Indon." with "Philip.", you will be going through the land regions in a counter clockwise way, with Indochina at the top, moving south to Sumatra, moving east to Java, followed by Borneo and E. Indonesia and finally, moving north to Philip- pines. Currently, it seems you are moving counter clockwise, then suddenly zipping to Philippines on top, before going back to E. Indonesia.

Figure 7: These panels are of great interest to the region. Although it is not possible to present everything within the manuscript, but the summer and winter plots for every land and sea region should be included in a supplement.

Figure 8 and 9: The authors mentioned that the FAIE signal maps are shaded if the FAIE signal is deemed statistically significant. The problem is when you allocate white (or a pale color) to 0 FAIE signal in the color bar, it is difficult to differentiate whether an area is shaded or not.

Acknowledged.

*Section 3.1.3: Where is West Pacific (Pg 13 Ln 9)? This is an example of figures that could be used more effectively.*
*I suppose the region to look at is the 3 grids at the top right hand corner of each panels (similar to West Pacific as*

*mentioned in Figure 7 bottom panel). If the boundaries of the 3 grids can be made bold, it will make it easier for the readers to follow. Pls identify similar issues with the use of figures within the manuscript and make changes. I have highlighted a few (pls see uploaded pdf).*

Thank you, the West Pacific region will be more specified in this section.

*Section 3.2.2: It is difficult for the readers to follow the impact of smoke on reff without visualizing the spatial distributions of smoke transport (or plumes) over SEA. It may be good to include NAAPS AOD plots for boreal summer and winter in the manuscript as well to orientate the readers to the direction of smoke transport.*

Thank you for the suggestion. We have added a figure showing the mean NAAPS AOD for each region and season.

*Figure 6: Pls consider swapping info for "E. Indon." with "Philip." When the reader go through the info, they would expect it to be presented in a logical manner. If you swap info for "E. Indon." with "Philip.", you will be going through the land regions in a counter clockwise way, with Indochina at the top, moving south to Sumatra, moving east to Java, followed by Borneo and E. Indonesia and finally, moving north to Philippines. Currently, it seems you are moving counter clockwise, then suddenly zipping to Philippines on top, before going back to E. Indonesia.*

I am unsure whether you are referring to Fig. 6 (baseline mean $r_{eff}$ maps) or Fig. 7 (line distributions of $r_{eff}$). No changes were made to acknowledge your comment.

*Figure 7: These panels are of great interest to the region. Although it is not possible to present everything within the manuscript, but the summer and winter plots for every land and sea region should be included in a supplement.*

Acknowledged. Distributions for all regions and seasons will be included in a supplement to the manuscript.

*Figure 8 and 9: The authors mentioned that the FAIE signal maps are shaded if the FAIE signal is deemed statistically significant. The problem is when you allocate white (or a pale color) to 0 FAIE signal in the color bar, it is difficult to differentiate whether an area is shaded or not.*

Thank you for the suggestion. The color bar was already modified to exclude pale colors. The authors believe the existing color scale in the manuscript provides enough contrast.

[revised manuscript text omitted]

The biases over land are larger than over ocean,

| Page 21: [2] Deleted | Alexa Ross | 7/5/18 4:36:00 PM |

–

| Page 21: [2] Deleted | Alexa Ross | 7/5/18 4:36:00 PM |

–

| Page 21: [2] Deleted | Alexa Ross | 7/5/18 4:36:00 PM |

–

| Page 21: [2] Deleted | Alexa Ross | 7/5/18 4:36:00 PM |

–

| Page 21: [2] Deleted | Alexa Ross | 7/5/18 4:36:00 PM |

–

| Page 21: [3] Deleted | ROBERT E HOLZ | 7/7/18 11:44:00 AM |

.

| Page 21: [3] Deleted | ROBERT E HOLZ | 7/7/18 11:44:00 AM |

.

| Page 21: [3] Deleted | ROBERT E HOLZ | 7/7/18 11:44:00 AM |

.

| Page 21: [4] Deleted | Alexa Ross | 7/7/18 1:12:00 PM |

| Page 21: [4] Deleted | Alexa Ross | 7/7/18 1:12:00 PM |

| Page 21: [5] Deleted | ROBERT E HOLZ | 7/7/18 11:45:00 AM |

these

| Page 21: [5] Deleted | ROBERT E HOLZ | 7/7/18 11:45:00 AM |

these

| Page 21: [5] Deleted | ROBERT E HOLZ | 7/7/18 11:45:00 AM |

these

| Page 21: [5] Deleted | ROBERT E HOLZ | 7/7/18 11:45:00 AM |

these

| Page 21: [5] Deleted | ROBERT E HOLZ | 7/7/18 11:45:00 AM |

these

| Page 21: [6] Deleted | Alexa Ross | 7/5/18 4:00:00 PM |

of the

| Page 21: [7] Deleted | Alexa Ross | 7/5/18 4:00:00 PM |

–

| Page 21: [8] Deleted | Alexa Ross | 7/5/18 4:01:00 PM |

ure 7

| Page 21: [8] Deleted | Alexa Ross | 7/5/18 4:01:00 PM |
|---|---|---|

ure 7

---

## Author Response (AR1)

Author comments on "Exploring the First Aerosol Indirect Effect over the Maritime Continent Using a Ten-Year Collocated MODIS, CALIOP, and Model Dataset" by Ross et al.

5

**Anonymous Referee #1 General Comments**

- This manuscript combined satellite observations and model reanalysis data (NAAPS) to study the relationship between aerosol and cloud effective radius (re) over Southeast Asia. They used both MODIS, CALIPSO, and model results to present the collocated dataset for analysis of aerosol-cloud relationship, which is critical for this kind of scientific topic. However, there have some fundamental issues need to be addresses carefully. First, they used aerosol information from the model simulations, since MODIS retrieval only provides few retrievals in moderate and high cloudy environment, and CALIPSO is unable to see aerosol below a cloud layer under some
- 15 conditions, as the authors stated. The model can indeed provide much more aerosol information, including spatial-temporal distribution, size distribution, chemical component, etc., but the method to collocate aerosol from model and cloud from satellite retrieval are unclear, especially the model results are 6 hourly products, and how to collocate with the time of satellite overpass? the model spatial resolution is much lower than satellite retrieval, how to collocate them in spatial? This need to be clarify. Second, it is unclear that how accurate is the
- 20 modeled aerosol compared to the satellite retrieval. The authors gave the statistical analysis of the AOD differences between the model and satellite, as well as the vertical profile, but what about the spatial distribution of the model results in comparison to satellite retrievals? Also, only the relationship between aerosol and cloud effective radius is studied, and definition of first aerosol indirect effect (FIRE) on p.15 and use of this definition in the title is inappropriate. The results and conclusions are not quite new. The anti-correlation of aerosol and
- 25 cloud effective radius over the oceanic regions have been reported extensively.

**Response to Anonymous Referee #1 General Comments**

- The referee comments expressed concern over the method of collocation between satellite retrievals and model output. The authors address the method of collocation in a simplistic manner: after a MODIS-CALIOP FOV is collocated, we simply find which 6-hourly 1x1-degree grid fits 'closest' in space and time. The model output from that 1x1-deg grid is then assigned to the FOV. This is also clearly illustrated in Figure 1. The referee's comments also addressed uncertainty within the modeled aerosol and the satellite observed aerosol. As stated in Ross et al., MODIS aerosol retrievals available are assimilated into the NAAPS reanalysis. However, due to the near
- 35 ubiquitous cloud cover in SEA, MODIS aerosol retrievals in the region are infrequent. Therefore, we provide AOD bias statistics between the observations and model to baseline relative model performance.

The referee is also implying that a higher resolution model would perhaps provide a better information to compare to satellite retrievals. However, over a large domain such as SE Asia the exact opposite is true. There 40 are numerous perturbations to the aerosol environment from a host of multi-scale processes, that cannot be reproduced in any model, nor even monitored from any available system. Rather, here we take more of a regime approach. When the model suggests above or below average particle concentrations, over all we find decreased and increased effective radius, respectively. In regions with low average concentrations, the effect is stronger

than regions with high average concentrations. Through such analyses, we can subsequently project regionally into climate change timescales. This leads to the second major comment on the *First Aerosol Indirect Effect*.

- We understand that the use of the term *First Aerosol Indirect Effect* is not precisely correct, since Ross et al. does 5 not explore any radiative aerosol indirect effect. Ross et al. is more a study of cloud-aerosol interactions than aerosol indirect effects. However, the anti-correlation of aerosol amount and droplet size (effective radius) is the exact mechanism through which the *FAIE* is theorized. We also argue that the results and conclusions are quite new. The paper shows that throughout Southeast Asia, effects of aerosols on maritime clouds have far reaching effects. Additionally, this paper shows that over land areas during seasons of biomass burning (such as Indochina
- 10 during winter months), distributions of relatively clean and relatively polluted cloud effective radii are extremely similar. This is in contrast to maritime regions (i.e. West Pacific winter), where relatively clean and relatively polluted clouds have very different effective radius distributions (Figure 7 panels a and c). The conclusions of this paper go beyond a simple anti-correlation between aerosol and effective radius, they provide detailed insight into cloud-aerosol interactions in an area of the world that is very hard to both observe and model.
- 15

**Responses to Anonymous Referee #1 Specific Comments**

p.3, bottom line, what aerosol amount? Please clarify.

As mentioned in the manuscript, the term 'aerosol amount' is used to infer NAAPS AOD. However, since 20 NAAPS AOD is determined using aerosol mass concentrations, assimilated with MODIS and MISR AOD, I thought the term aerosol amount was appropriate.

*p.4, line 12-15, the study of Ma et al. (2010) was cited in the text, but not listed in reference.* Thank you for pointing this out, I will add Ma et al. to the references.

25

45

Section 2.3, p. 6, line 9 15, it is unclear that how to collocate the data, e.g. line 19-10, level 1 or level 2, or both? Also, on the collocation of time. As described in sec. 2.2 (p.5), the model results are 6-hourly product, so how to collocate the model results with the time of satellite overpasses?

- These concerns were addressed in the response to general comments above. Section 2.3.1 further explains the 30 collocation process between satellite observations and model output. Regarding the matching in time, once the collocation between MODIS and CALIOP is performed, that FOV has an assigned overpass time (MODIS timestamp). We simply find which 6-hourly model field is closest in time to the MODIS timestamp. I will add this clarification into section 2.3.1.
- 35 p.8, line line 5-9, what is the average of AOD over the region?We are unsure what your comment is referring to, but I believe it is p. 9 in section 2.4. I will refer you to table 3.1 where we list median of AOD for each region and season in this paper.

Figure. 5, top panel is for above sea level, not surface. Thank you for pointing out this typo,

40 I will correct the y-axis text in Figure 5.

p.12, line 1-2, the definition of aerosol anomaly is difficult to understand. 'when the AOD is greater than an upper threshold, ( $\mu AOD \times \sigma AOD$ ). A negative aerosol anomaly, or clean event, is when the AOD is less than a lower threshold, ( $\mu AOD / \sigma AOD$ )', why the polluted and clean condition are defined like this? The aerosol anomalies are defined as such because what is anomalous for one region/season isn't necessarily anomalous for

another. Upper and lower thresholds are defined for each region/season because what is considered 'clean' in a polluted land region like Indochina should not also be considered 'clean' in a maritime region like the West Pacific. Average AOD and therefore aerosol anomalies must be defined individually for each region/season.

5 Figure. 7: . . . in the lower left", should be lower right.

Thank you for pointing this out, I will change the caption text in Figure 7.

p.15, top of the page, the definition of FAIE. This definition is only the differences of cloud effective radius between the polluted (clean) conditions and baseline. This is actually not FAIE, so the definition is incorrect and 10 misleading.

You are correct in pointing out that the full definition of the FAIE includes an albedo effect. In this paper, we only discuss the mechanism which drives the FAIE, namely the anti-correlation between effective radius and AOD. I addressed these concerns in the response to general comments. We will therefore leave the FAIE terminology in the paper.

15

p. 17, line 7: aerosol amount, what aerosol properties is used in the analysis, AOD or aerosol number? or something else?

As mentioned in Section 2.4, aerosol amount is determined by NAAPS AOD.

p.17, line 10-11, finding..., this is not new. 20

We hold that aerosol effects from land regions in SEA do have far reaching effects on marine clouds. We believe this result to be original and new. We have addressed these concerns in the response to the general comments.

**Anonymous Referee #3 General Comments**

25

1) Comprehensive literature review on related studies in SEA (besides classics on aerosol effects and authors' publications) is needed to put the work in context. E.g., Lin et al. (2014) gave an overview on interactions between biomass burning aerosols and clouds over SEA, and Lee et al. (2014) modelled the impact of aerosols on atmospheric circulation and rainfall over SEA.

30

2) I think the structure of the paper is slightly skewed at the moment. The paper devotes almost half of the pages to introduction and methods, and the other half to results. Given that the paper describes a new dataset, a comprehensive coverage to the methods is quite understandable. That being said, the discussions on the results are thus comparatively limited. I think more discussions of these results in relation to the biomass burning and pollution situation in SEA are possible. Pls see specific comments for more details.

35

3) I think the authors can make better use of the figures to expand the discussions. E.g., panels in the figures can be labelled "a, b, c ...". Firstly, this will help the readers to relate the in-text discussions to the different panels in the figures or features within the panels. Secondly, the authors may also find it easier to focus on specific panels

- 40 to highlight point of interests. Pls see specific comments for more details. E.g., the punchline of the paper in my opinion is Figure 8 and 9 and are discussed in only two pages out of 17 pages of text, focusing mainly on the statistics on what made the FAIE signal significant. There are maybe two paragraphs (less than a page) on how these results relate to biomass burning and pollution over the region. Moreover, these two figures are rather complicated for the uninitiated, and consists spatial, temporal and vertical info for 6 land regions and 20 sea
- 45 regions (4 aggregated sea regions). The authors should expand the discussions to take the readers through the

richness of these two figures. Are there opportunities to explore how the variations of injection heights of smoke and pollution impact the cloud reff at each level? E.g. look carefully at Sumatra and Java for Figure 9. reff for Sumatra gets warmer (decreases) downwards vertically, but reff for Java seems be warmest (lowest reff) at 3 - 4.5 km. Perhaps the authors can explore if this be due to convective pumping bringing the pollution (not smoke) to

5 that level? Do note that these are suggestions to take the discussion further in more details.

4) The authors should refer to ACP manuscript preparation guidelines for authors (at https://www.atmosphericchemistry-and- physics.net/for\_authors/manuscript\_preparation.html) when preparing manuscript. E.g. Pg 6 Ln 25: "as illustrated in Figure 1" should be "as illustrated in Fig. 1" according to guidelines. Another e.g. within the same line: "333-m" should be "333 m" according to house standard not to hyphenate modifiers containing

abbreviated units. The authors should also check the language within the manuscript (some suggestions are given in the uploaded pdf. There are also numerous uses of semicolons which break up the flow in the text (pls see uploaded pdf as well).

15

10

**Response to Anonymous Referee #3 General Comments**

We appreciate your thorough comments on our manuscript. We have incorporated some of your citation recommendations. The significance of the results were slightly expanded upon in Sections 3 and 4 of the paper,
though we felt we got most of our points across concerning the anti-correlation of reff and AOD and how the anti-correlation related to seasons of biomass burning. The updated manuscript includes more discussion of the key figures (now Figs. 8, 9, and 10). We have also added a mean AOD figure for each season/reason, as per your suggestion (Fig. 6). Thank you for the corrections on the ACP manuscript preparation guidelines. We made the necessary corrections. We have also adopted most of your suggestions in the supplemental draft.

25

**Responses to Anonymous Referee #3 Specific Comments**

*Title: The entire SEA is effectively covered in the study. It will be appropriate to title the manuscript as such.* Maritime Continent will be changed to Southeast Asia in the title of the paper.

30

Abstract: You may want to introduce CCARA here to the scientific community. This is important as interested readers will pick it up immediately and possibly relate to future work using the dataset. You don't want CCARA to be lost in the rest of the text.

CCARA will be introduced in the abstract.

Pg 2 Ln 17: I have not seen this term "Greater Southeast Asia" being used in literature. Acknowledged. Changed to 'Southeast Asia.'

Pg 6 Ln 1 - 3: I understand that ECMWF met data performs well for SEA, but it is equally important that NAVGEM performs well too as it is used to drive aerosol transport used in the study. The authors should provide evidence that the mismatch between met and modelled aerosol parameters is acceptable (e.g., statistics from the data) or only the mentioned key parameters of e.g., wind shear, water vapor, are important for the phenomenon being studied (e.g., citations).

Technically both are on the dataset, but because of moisture and stability parameters we focus on ECMWF meteorology to provide the best available state vector to describe the cloud environment. The NAAPS model, as driven by NAVGEM does indeed have different meteorology, but internal studies show wind patterns are quite consistent, especially in the analyses which is utilized here. The only place where this would lead to a significant

- 5 difference is in precipitation for wet scavenging. But here, recognizing the wide diversity in precipitation between models, NAAPS utilizes satellite based CMORPH precipitation which was shown in Xian et al., 2010 to markedly improve skill. Thus, the purpose of the ECMWF meteorology and the meteorology driving the NAAPS model and data assimilation system serve fundamentally different purposes. Ultimately, from an aerosol point of view, NAAPS aerosol skill and CAMS/ECMWF and GEOS 5 skill are comparable in the region (See Sessions et
- 10 al., 2015). If anything, NAAPS reanalysis with its use of satellite precipitation currently edges-out other models.

Section 3.1.1: Injection heights of biomass burning smoke is extremely important for determining the vertical distribution of biomass burning aerosols. How is the injection height in NAAPS determined? Does the method matter for the different varieties of "biomass burning" present in SEA (Reid et al., 2013)?

- 15 For SEA, biomass burning emissions does not seemed to be directly injected into the middle to upper troposphere (Campbell et al., 2013; Tosca et al., 2011). As mentioned by Reid et al. (2013), the type of "biomass burning" in SEA actually varies a lot, thus possibly resulting in a wide range of injection heights. The authors should discuss the bias for CALIOP-NAAPS in more details framed in the context of the above-mentioned papers. The model results seem to be higher than the CALIPSO results, especially for ocean (Fig. 5).
- 20

Pg 8 Ln 9 - 16: The four larger ocean grids (e.g., ocean - south west) mentioned here should be named. They are later mentioned in Table 1 without reference back to Figure 3. A single sentence in Section 3.1.2 like e.g., "the north west, north east, south west and south east of ocean mentioned in Table 1 are with reference to the larger ocean grids (Fig. 3a)." will help the readers much in understanding the analysis. Acknowledged, the four larger ocean grids will be identified earlier here. Clarification will also be added in Section 3.1.2.

Pg 11 Ln 4 and 9: Campbell et al. (2012) are cited in text while referenced as Campbell et al. (2013) in reference section. Pls kindly check through text for citation errors. Acknowledged.

30

35

25

Section 3.1.3: Where is West Pacific (Pg 13 Ln 9)? This is an example of figures that could be used more effectively. I suppose the region to look at is the 3 grids at the top right hand corner of each panels (similar to West Pacific as mentioned in Figure 7 bottom panel). If the boundaries of the 3 grids can be made bold, it will make it easier for the readers to follow. Pls identify similar issues with the use of figures within the manuscript and make changes. I have highlighted a few (pls see uploaded pdf).

Thank you, the West Pacific region will be more specified in this section.

Section 3.2.2: It is difficult for the readers to follow the impact of smoke on reff without visualizing the spatial distributions of smoke transport (or plumes) over SEA. It may be good to include NAAPS AOD plots for boreal
 summer and winter in the manuscript as well to orientate the readers to the direction of smoke transport.

Thank you for the suggestion. We have added a figure showing the mean NAAPS AOD for each region and season.

Figure 6: Pls consider swapping info for "E. Indon." with "Philip." When the reader go through the info, they would expect it to be presented in a logical manner. If you swap info for "E. Indon." with "Philip.", you will be

going through the land regions in a counter clockwise way, with Indochina at the top, moving south to Sumatra, moving east to Java, followed by Borneo and E. Indonesia and finally, moving north to Philippines. Currently, it seems you are moving counter clockwise, then suddenly zipping to Philippines on top, before going back to E. Indonesia.

5 I am unsure whether you are referring to Fig. 6 (baseline mean  $r_{eff}$  maps) or Fig. 7 (line distributions of  $r_{eff}$ ). No changes were made to acknowledge your comment.

Figure 7: These panels are of great interest to the region. Although it is not possible to present everything within the manuscript, but the summer and winter plots for every land and sea region should be included in a supplement.

Acknowledged. Distributions for all regions and seasons will be included in a supplement to the manuscript.

Figure 8 and 9: The authors mentioned that the FAIE signal maps are shaded if the FAIE signal is deemed statistically significant. The problem is when you allocate white (or a pale color) to 0 FAIE signal in the color bar, it is difficult to differentiate whether an area is shaded or not.

Thank you for the suggestion. The colorbar was already modified to exclude pale colors. The authors believe the existing colors in the manuscript to provide enough contrast.

[revised manuscript text omitted]

aerosol-precipitation relationships will be limited to a five-year time period. For the research presented in this manuscript, we utilized the entire collocated MODIS-CALIOP record from 2006-2016.

The focus of this analysis is to investigate a cloud-aerosol relationship known as the first aerosol indirect effect (or FAIE), introduced by S. Twomey in 1974, with the premise that for aerosol particles to affect precipitation in isolated warm

- 5 cumulus clouds, they likely have to first influence cloud effective radius (reff). Using the curtain dataset described herein, we first explore the observational challenges of the aerosol and warm convective cloud system, followed by an analysis that includes the use of Navy Aerosol Analysis and Prediction System (or NAAPS; Lynch et al., 2016) reanalysis for aerosol detection. Indeed, model aerosol data has been combined with satellite retrieved properties before. Ma et al. (2010) uses aerosol concentrations simulated by the GOCART model to show a dependency of observed cloud droplet size provided by
- 10 MODIS on organic carbon amounts. The study by Ma et al. (2010) was for a global domain and did not investigate the long-reaching effects of aerosol particles or the scale of the impact of pollution on cloud droplet size. Additionally, Ma et al. (2010) did not investigate the dependency of the FAIE on cloud top height, which can be used as a proxy for filtering different types of clouds.

This paper focuses on the MODIS, CALIOP and NAAPS components of the Southeast Asian Curtain Cloud-Aerosol 15 Regional A-Train (CCARA) dataset, and structured in the following way. In Sect. 2, we describe the generation of the A-

Train curtain dataset. In Sect. 3, we examine the baseline statistical properties and quality assurance of the products therein, with a particular emphasis on sampling and the potential for thin cirrus cloud contamination. Sect. 4 provides results and discussions on observed relationships between aerosol and cloud  $r_{effe}$ . A final discussion and conclusion synopsis is provided in Sect. 5.

**20 2 Methods: Development of the Southeast Asian Curtain Cloud-Aerosol Regional A-Train (CCARA) product**

The dataset used here was developed specifically to meet the challenges of doing aerosol and cloud research in the complex SEA environment. Here, we demonstrate this system at its simplest level as a launching point for more detailed analysis of the CCARA. The dataset is comprised of data from two satellite instruments and fields from two models. In short, the basis is CALIOP providing cirrus screening and cloud height information. Other cloud properties are derived from MODIS.

25 Aerosol properties are derived from the NAAPS-RA (which includes aerosol optical depth (AOD) from MODIS and the Multi-angle Imaging Spectroradiometer (MISR) instrument AOD, with supporting meteorological information from the ERA-Interim atmospheric reanalysis (or European Reanalysis dataset) (Dee et al., 2011) produced by the European Centre for Medium-Range Weather Forecasts (ECMWF),

Sect. 2.1 includes details of the satellite instruments. Sect. 2.2 describes the model fields. Sect. 2.3 explains the technique by

30 which the satellites and model fields were collocated to build CCARA.

**2.1 Description of the satellite instruments**

MODIS flies on board the Aqua satellite, which launched in 2002 (Salomonson et al., 2002). The imager observes reflectance at 36 wavelengths in the visible and infrared bands and spatial resolutions ranging from 250 meters to 1 km. The

10

| Deleted: observing |
|--------------------|
|--------------------|

|
|--------------|-----|
|              |     |

| -  | Deleted: Section    |
|----|---------------------|
| -{ | Deleted: Section    |
| -{ | Deleted: Section    |
| -{ | Deleted: properties |
| -{ | Deleted: Section    |

| Deleted: the               |  |  |
|----------------------------|--|--|
| Deleted: footprint         |  |  |
| Deleted: to provide        |  |  |
| Deleted: data assimilation |  |  |

MODIS products used in this research are from Level 2 Collection 6 (C6), provided at spatial resolutions spanning 1 to 10 km2 at nadir. In this study, the primary MODIS data used are cloud property products (MYD06; Platnick et al., 2017), the cloud mask products (MYD35; Ackerman et al., 1998) and aerosol product (MYD04, Levy et al., 2005). MODIS collocation indices are found using the MYD03 geolocation files at 1 km spatial resolution. Partly cloudy (PCL) pixels are classified in 5 the C6 MYD06 algorithm, but are not included in this analysis.

The CALIOP instrument provides observations at two wavelengths: 532 nm (visible) and 1064 nm (near-infrared) with polarization capabilities in the 532 nm channel. The lidar has a vertical resolution of 30 m, producing detailed profiles of cloud and aerosol vertical structure. CALIOP Level 2 cloud products exist at horizontal resolutions of 333 m, 1 km, and 5 km. CALIOP Level 2 aerosol products exist at 5 km horizontal resolution. All of the CALIOP data used in this analysis are

10 from the 5 km cloud and aerosol products (CPRO and APRO) which while having 60 m vertical resolution, can include spatial averages up to 80 km depending on the signal-to-noise ratio. CALIOP Level 2 algorithms provide their own set of feature layers that can distinguish up to 10 cloud and aerosol layers (CLAY and ALAY; Winker et al., 2009). To further take advantage of the CALIOP-resolved profile without having to save the full CALIOP profiles, customized layers defined to integrate with NAAPS vertical layers were designed specifically for this research and are discussed in Sect. 2.3.2. We use 15 CALIOP aerosol products to verify the NAAPS reanalysis, and CALIOP cloud products to define cloud top heights and screen for thin cirrus.

**2.2 Description of model products**

The NAAPS reanalysis (Lynch et. al., 2016) is a decade-long (2003-2016) global 1x1 degree and 6-hourly aerosol reanalysis product, which was developed and validated at the Naval Research Laboratory. This reanalysis utilizes a modified version of

- 20 NAAPS as its core and assimilates quality-controlled retrievals of AOD from MODIS on Terra and Aqua and MISR on Terra (Zhang et al., 2006; Hyer et al., 2011; Shi et al., 2014). NAAPS characterizes anthropogenic and biogenic fine (including sulfate, and primary and secondary organic aerosols), dust, biomass burning smoke and sea salt aerosols. Smoke from biomass burning is derived from near-real-time satellite-based thermal anomaly data to construct smoke source functions (Reid et al., 2009), with additional orbital corrections on MODIS-based emissions and regional tunings to mitigate
- 25 missing fire detection resulting from daily variations in orbital coverage and cloud obscuration. Aerosol wet deposition in the tropics is driven by NOAA Climate Prediction Center (CPC) MORPHing (CMORPH) precipitation derived from satellite observations (Joyce et al., 2004) to correct model precipitation biases that ubiquitously exist in numerical models (Xian et al., 2009). The 3-dimensional aerosol fields are resolved vertically into 25 layers between the surface and about 70\_mb following a sigma-pressure coordinate.
- 30 In addition to NAAPS, the CCARA dataset includes ERA-Interim atmospheric reanalysis. Meteorological fields such as pressure, temperature, specific and relative humidity, winds, and divergence are included with the CCARA product. Although not used specifically in this paper on simple effective radius sensitivity, the ERA-Interim reanalysis provides important information on the cloud regional environment. While NAAPS derives its meteorology from the Navy Global

[revised manuscript text omitted]

profiles with both the cloud and aerosol extinctions retrieved at 30 m vertical resolution. To facilitate direct comparison between the model fields and the CALIOP cloud and aerosol products, we have defined two sets of customized vertical layers for CCARA: one that captures the full vertical column at 1 km vertical resolution, and one at slightly higher resolution that captures the planetary boundary layer (PBL).

- 5 The first set of layers that captures the full atmospheric column is defined with respect to the mean sea level (MSL). They parse the vertical dimension from 0-18 km in 1 km increments (each layer is 1 km in thickness). These are referred to as MSL layers. The second set of layers has a finer vertical resolution and is defined with respect to the surface elevation; these nine layers are referred to as PBL layers. The nine PBL layers, defined with respect to surface elevation, are: 200-500 meters, 0.5-1 km, 1-1.5 km, 1.5-2 km, 2-2.5 km, 2-5-3 km, 3-4 km, 4-5 km, and 5-6 km. The two sets of layers are illustrated
- 10 in Fig. 2. We do not include a 0-200 meters layer because the effects of surface reflectance on CALIOP retrievals (Winker et al., 2009).

For each layer in the MSL and PBL layer sets, NAAPS extinctions and aerosol mass concentrations are interpolated and saved. These are separately attributed to four aerosol species anthropogenic and biogenic fine mode (or ABF), dust, smoke, and sea salt, and to two aerosol modes, fine and coarse. The CALIOP aerosol and cloud extinction profiles are integrated

15 within each layer providing the layer optical depth. The fractions of cloud or aerosol detected within the layer by CALIOP are also saved. These CALIOP-derived layer products are calculated using the Version 3 Level 2 profile products which provide aerosol and cloud extinctions at 30 m resolution. Vertically-resolved ERA-interim meteorology fields (the European Reanalysis Interim data) are also included for the PBL and MSL layers.

**2.4 Analytical techniques**

- 20 This section details the methods of analysis and describes the definition of the First Aerosol Indirect Effect (FAIE) using the CCARA dataset. While the dataset has many dimensions of aerosol, cloud and meteorological parameters, for this first introductory paper we are looking at the simple AOD-cloud comparisons, accounting for the deep\_geated regional biases present in nearly all datasets investigating Southeast Asia (Reid et al., 2013). Future papers will further explore aerosol-cloud indirect effects in SEA.
- 25 Input intercomparison: This analysis aims to evaluate the extent to which aerosol particle concentrations and type impact the cloud properties (namely cloud effective radius) of liquid clouds. To this end, we cross-correlated modeled aerosol optical depth anomalies (provided by NAAPS) with cloud reff (observed by MODIS), using CALIOP to filter for cirrus contamination and provide the cloud top height. As an initial validation of this approach, we first intercompare the total columnar AOD between satellite observations and the NAAPS over land and ocean, AOD innovation is the observed AOD
- 30 minus the model-derived AOD, or (AODobstAODNAAPS), where the observation can be CALIOP or MODIS. For ocean retrievals, AOD innovations average 0.02 for MODIS-NAAPS and -0.05 for CALIOP-NAAPS. Over land, MODIS-NAAPS innovation has an average of 0.05 compared to 0.07 for CALIOP-NAAPS.

|   | 4 |
|---|---|
| • |   |

| Deleted: | -            |
|----------|--------------|
|          |              |
| Deleted: | -            |
| Deleted: | 500 meters - |
| Deleted: | -            |
| Deleted: | -            |
| Deleted: | -            |
| Deleted: | -            |
| Deleted: | -            |
| Deleted: | -            |
| Deleted: | -            |
| Deleted: | Figure       |
| Deleted: | -            |
| Deleted: | hinder the   |
| Deleted: | separated by |
| Deleted: | ,            |
| Deleted: | by           |
| Deleted: | ,            |
| Deleted: | or           |
| Deleted: | v            |
| Deleted: | -            |
| Deleted: | signal       |
| Deleted: |              |

| -{ | Deleted: -                                         |
|----|----------------------------------------------------|
| -  | Deleted: , separating by land and ocean retrievals |
| -{ | Deleted:                                           |
| 1  | Deleted: -                                         |

Partitioning the region: SEA is partitioned into oceanic and land sub-regions as we hypothesize different regions of the domain will have varying sensitivities to the FAIE. Ocean grids and land regions are shown in Fig. 3. Ocean retrievals are divided into twenty 10x10-degree grids to analyse the FAIE (Fig. 3 right panel). To compute thresholds for ocean aerosol anomalies, we use four larger ocean grids to reduce the uncertainty in the anomaly calculation (Fig. 3 left panel). Land

- 5 retrievals are divided into six regions which are: 1) Peninsular Southeast Asia (Myanmar, Thailand, Cambodia, Laos, Vietnam), 2) Sumatra and peninsular Malaysia, 3) Borneo, 4) Java, 5) the Philippines, and 6) The eastern domain of Sulawesi, Bali, Timor, and New Guinea (hereby East Indonesia). Land aerosol anomaly thresholds are calculated using the same six land regions.
- *Aerosol anomalies*: Like all global aerosol models, it is expected that the NAAPS model has varying regional efficacy. Yet, global models can be expected to simulate large perturbations for large-scale aerosol features (e.g., Sessions et al., 2015), but
- finer scale local and mesoscale effects with poorer fidelity. Thus, direct comparisons between the global aerosol model and  $r_{eff}$  can lead to regionally varying biases, which in turn can cloud the impact of varying aerosol concentrations on cloud properties. However, perturbations in the large-scale model should reflect regional changes associated with significant biomass burning or anthropogenic pollution events. Thus, correlating NAAPS aerosol perturbations to cloud properties can
- 15 isolate how these large events can influence the cloud environment. Anomalous aerosol amounts were defined by examining the full column fine mode AOD from NAAPS. A scene is determined to be anomalous if the AOD is more or less than an average AOD by one standard deviation. The AOD anomaly thresholds were calculated using the ten-year NAAPS AOD that have been mapped into the CCARA product. Distributions of aerosol mass load were divided by species, geographic region, surface type, and season. For each ocean grid and land region, ten-year distributions of aerosol load (integrated)
- 20 aerosol mass concentration) are found separating boreal winter and summer months (i.e. winter is December through May and summer is June through November).
  Aerosol thresholds: Our analysis uses the NAAPS fine mode AOD anomalies. Fine mode AOD is used as opposed to total or

speciated AOD to encompass any possible aerosol acting as CCN. Distributions of NAAPS AOD that are collocated into the CCARA dataset are compiled. A mean AOD ( $\mu_{AOD}$ ) and standard deviation of AOD ( $\sigma_{AOD}$ ) are calculated for each region

25 and season. We assume distributions of AOD are log-normal, thus  $\mu_{AOD}$  and  $\sigma_{AOD}$  are the geometric mean and geometric standard deviation of an AOD population. A pixel with NAAPS AOD < ( $\mu_{AOD} \pm \sigma_{AOD}$ ) is considered a negative anomalous aerosol event (clean event); AOD > ( $\mu_{AOD} \pm \sigma_{AOD}$ ) is a positive anomalous aerosol event (polluted event). This threshold is defined such that for each FOV in the CCARA dataset, there is an assignment of more/less/null aerosol amount for each of the four aerosol species and each of the two aerosol modes. Understandably, different regions experience different clean or

30 polluted seasons due to seasonal and regional patterns of biomass burning and land use change (Xian et al., 2013; Reid et al., 2013), Thus, we define aerosol anomaly as relative to where and when a retrieval is.

*Filtering*: Distributions of MODIS liquid reff can be partitioned using a range of criteria such as region and surface type (land or ocean), season, cloud top height, and aerosol anomaly. To identify correlations, bulk 'baseline' distributions are calculated that do not filter by any AOD threshold; baseline distributions are meant to serve as a benchmark or control to compare with

15

| Deleted: ;         Deleted: ;         Deleted: Ocean analysis grids and ocean aerosol grids and lan regions are shown in Figure 3.         Deleted: ocean analysis grids and ocean aerosol grids and lan regions are shown in Figure 3.         Deleted: with poorer fidelity for capturing         Deleted: eross         Deleted: aerosol optical thicknesses         Deleted: a arosol optical thicknesses         Deleted: s         Deleted: nd saved         Deleted: *         Deleted: efinition | ·       |                                      |
|----------------------------------------------------------------------------------------------------------------------------------------------------------------------------------------------------------------------------------------------------------------------------------------------------------------------------------------------------------------------------------------------------------------------------------------------------------------------------------------------------------|---------|--------------------------------------|
| Deleted: ;
regions are shown in Figure 3.
| Deleted: ;
regions are shown in Figure 3.
| Deleted: ;
regions are shown in Figure 3.
| Deleted: ;
regions are shown in Figure 3.
Deleted: *                                                                                                                                                               | Doloto  | An -                                 |
| Deleted: ;
regions are shown in Figure 3.
Deleted: *                                                                                                                                                               | Delete  | 4                                    |
| Deleted: ,
regions are shown in Figure 3.
4                             |
| Deleted: Ocean anarysis grids and ocean derosor grids and ian regions are shown in Figure 3. Deleted: with poorer fidelity for capturing Deleted: cross Deleted: aerosol optical thicknesses Deleted: a and saved Deleted: s Deleted: / Deleted: * Deleted: aerosol optical thicknesses                                                                                                                                                                                                                  | Delete  | u: ;
d. O                         |
| Deleted: with poorer fidelity for capturing Deleted: cross Deleted: aerosol optical thicknesses Deleted: s Deleted: s Deleted: / Deleted: / Deleted: * Deleted: aerosol optical thicknesses                                                                                                                                                                                                                                                                                                              | regions | are shown in Figure 3.               |
| Deleted: with poorer fidelity for capturing Deleted: cross Deleted: aerosol optical thicknesses Deleted: s Deleted: s Deleted: / Deleted: / Deleted: * Deleted: efinition                                                                                                                                                                                                                                                                                                                                | -       |                                      |
| Deleted: aerosol optical thicknesses Deleted: s Deleted: ard saved Deleted: / Deleted: * Deleted: * Deleted: definition                                                                                                                                                                                                                                                                                                                                                                                  | Doloto  | d with poorer fidelity for cepturing |
| Deleted: cross Deleted: acrosol optical thicknesses Deleted: s Deleted: s Deleted: / Deleted: / Deleted: * Deleted: acrosol optical thicknesses                                                                                                                                                                                                                                                                                                                                                          | Delete  | a. with pooler indenty for capturing |
| Deleted: cross Deleted: acrosol optical thicknesses Deleted: s Deleted: and saved Deleted: / Deleted: * Deleted: acrosol optical thicknesses                                                                                                                                                                                                                                                                                                                                                             |         |                                      |
| Deleted: cross Deleted: acrosol optical thicknesses Deleted: s Deleted: s Deleted: / Deleted: / Deleted: * Deleted: acrosol optical thicknesses                                                                                                                                                                                                                                                                                                                                                          |         |                                      |
| Deleted: acrosol optical thicknesses Deleted: s Deleted: and saved Deleted: / Deleted: * Deleted: acrosol optical thicknesses                                                                                                                                                                                                                                                                                                                                                                            | Delete  | d: cross                             |
| Deleted: aerosol optical thicknesses Deleted: s Deleted: ad saved Deleted: / Deleted: * Deleted: * Deleted: ad saved                                                                                                                                                                                                                                                                                                                                                                                     |         |                                      |
| Deleted: aerosol optical thicknesses Deleted: s Deleted: and saved Deleted: / Deleted: * Deleted: efinition                                                                                                                                                                                                                                                                                                                                                                                              |         |                                      |
| Deleted: acrosol optical thicknesses Deleted: s Deleted: and saved Deleted: / Deleted: * Deleted: definition                                                                                                                                                                                                                                                                                                                                                                                             |         |                                      |
| Deleted: aerosol optical thicknesses
Deleted: definition                                                                                                                                                                                                                                                                                                                                                                              |         |                                      |
| Deleted: s
Deleted: efinition                                                                                                                                                                                                                                                                                                                                                                                                                       | Delete  | d: aerosol optical thicknesses       |
| Deleted: s Deleted: and saved Deleted: / Deleted: * Deleted: definition                                                                                                                                                                                                                                                                                                                                                                                                                                  |         |                                      |
| Deleted: s Deleted: and saved Deleted: / Deleted: * Deleted: efinition                                                                                                                                                                                                                                                                                                                                                                                                                                   |         |                                      |
| Deleted: s Deleted: and saved Deleted: / Deleted: * Deleted: efinition                                                                                                                                                                                                                                                                                                                                                                                                                                   |         |                                      |
| Deleted: s Deleted: and saved Deleted: / Deleted: * Deleted: definition                                                                                                                                                                                                                                                                                                                                                                                                                                  |         |                                      |
| Deleted: s Deleted: and saved Deleted: / Deleted: * Deleted: definition                                                                                                                                                                                                                                                                                                                                                                                                                                  |         |                                      |
| Deleted: and saved Deleted: / Deleted: * Deleted: efinition                                                                                                                                                                                                                                                                                                                                                                                                                                              | Delete  | d: s                          |
| Deleted: /
Deleted: definition                                                                                                                                                                                                                                                                                                                                                                                                                                                          | Delete  | d: and saved                         |
| Deleted: / Deleted: * Deleted: definition                                                                                                                                                                                                                                                                                                                                                                                                                                                                |         |                                      |
| Deleted: * Deleted: * Deleted: definition                                                                                                                                                                                                                                                                                                                                                                                                                                                                | Delete  | d• /                                 |
| Deleted: definition                                                                                                                                                                                                                                                                                                                                                                                                                                                                                      | Delete  | 4 = /
                            |
| Deleted: definition                                                                                                                                                                                                                                                                                                                                                                                                                                                                                      | Delete  | d: *                                 |
|                                                                                                                                                                                                                                                                                                                                                                                                                                                                                                          | Delete  | a: definition                        |
|                                                                                                                                                                                                                                                                                                                                                                                                                                                                                                          |         |                                      |

**Deleted:**

aerosol anomalous distributions. Cloud top heights are defined using the CALIOP cloud layer product and are divided into three regimes which include layers between 1-3 km (small cumulus), 3-4.5 km (cumulus mediocris), and 4.5-6 km (smaller cumulus congestus), defined above surface elevation. Typically, at 6 km, temperatures are at approximately -5°C, ensuring that clouds are likely not experiencing ice physics. However, clouds at this temperature may be mixed in phase. We use a 6

- 5 km height cut-off and liquid-only MODIS retrievals to increase the certainty that we are looking only at liquid water reff retrievals. CALIOP Level 2 cloud layer products are also used to filter FOV with thin ice cirrus contamination above the liquid cloud layer (Winker et al., 2009). From these filtered distributions, averages (geometric means) of reff for baseline, polluted (positive aerosol anomaly), and clean (negative aerosol anomaly) scenarios are calculated. Our goal is to investigate the FAIE by comparing the MODIS reff for polluted versus clean data to the baseline mean reff. The polluted/clean
- displacement of  $r_{eff}$  from the baseline  $r_{eff}$  is called the FAIE signal. Sampling bias and statistical significance: CALIOP is a nadir-viewing instrument in a polar sun-synchronous orbit. Even with a ten-year record, the sampling statistics when accumulated over small regions for a subset of clouds can be poor. For example, summertime data over the Philippines for land surfaces and clouds existing between 1-3 km yield only 1005 collocated MODIS and CALIOP reff retrievals. This limited sampling results not just from the nadir viewing geometry, but
- 15 also from the extensive cirrus cloud cover in the region that attenuates the CALIOP observations and limits the low-level cloud MODIS retrieval yield. After applying an additional filter for investigating the aerosol anomaly, only 40 of those retrievals occur during a positive fine mode aerosol anomaly. For comparison, the 1-3 km summertime clouds in the northernmost and easternmost 1-degree ocean grid have 8603 baseline reff and 461 retrievals after filtering for positive fine mode aerosol anomaly due to the larger surface area of this region and much higher frequency of low clouds over this ocean 20 grid compared to over the Philippines.

Statistical significance was determined by performing a two-sample t-test on each partitioned distribution of MODIS refr. The average clean and polluted reff are compared to the baseline reff. p-values are calculated and interpreted to give the certainty that the mean of an anomalous distribution is significantly different from the mean of a baseline distribution. We also specify the type of alternative hypothesis that is tested in the t-test, meaning we can determine to what certainty the mean of an

25 anomalous distribution is more or less than the mean of a baseline distribution. Thus, we can answer specifically whether smaller cloud droplets are more likely observed in a polluted environment compared to the baseline (or "normal") environments. Similarly, we test the likelihood that larger mean reff occur in a clean environment compared to the normal environments. We consider any p-value less than 0.01 to signify a significant difference between polluted/clean and baseline distributions.

**30 3 Results**

10

This section focuses on results from the ten-year CCARA dataset over Southeast Asia that includes observations from MODIS and CALIOP as well as NAAPS and ERA-interim reanalysis. Sect. 3.1 explores the basic properties of the CCARA dataset, including intercomparisons between NAAPS aerosol reanalysis and the CALIOP and MODIS aerosol observational

| • |   |
|---|---|
| н | 6 |
| r | L |

| Deleted: – |  |  |
|------------|--|--|
| Deleted: - |  |  |
| Deleted: – |  |  |
| B.I.I.I    |  |  |
| Deleted: - |  |  |
|            |  |  |
|            |  |  |
|            |  |  |

| Deleted: S  | lection   |
|-------------|-----------|
| Deleted: re | eanalysis |

 $\frac{data}{data} \text{Then, we show the aerosol anomaly thresholds for each region and season, defined using the NAAPS fine mode AOD.} We also show the overall mean (or baseline) MODIS effective radius (reff) for each region and season without considering aerosol anomalies.$

Sect. 3.2 investigates the FAIE signal. To detect FAIE, we compare populations of baseline reff to clean and polluted reff.
 Statistical significance testing is performed on these populations to determine whether the mean of clean or polluted reff is greater or less than the mean of the baseline reff within degree of certainty. To pinpoint when and where the FAIE occurs, populations of reff are filtered by region (twenty ocean grids and six land regions) and season (winter or summer). Ocean and land regions are shown in Fig. 3. To further narrow the occurrence of FAIE, we filter by other fields in the CCARA dataset such as CALIOP cloud top height.

**10 3.1 Basic data properties**

This section details the intercomparison of the NAAPS, MODIS, and CALIOP AOD in the ten-year CCARA dataset over Southeast Asia. Here we explore AOD intercomparisons between NAAPS reanalysis and CALIOP and MODIS aerosol observations. We also explore average AOD in each region and season, which we use to define aerosol anomaly thresholds. Lastly, we examine the mean effective radius in each region and season, which we use later in Sect. 3.2 to explore the FAIE.

**15 3.1.1 Aerosol detection intercomparison: NAAPS versus the observations**

An AOD intercomparison between NAAPS and the observing platforms CALIOP and MODIS is presented in Fig. 4, showing 2-D histograms of AODMODISAODNAAPS and AODCALIOPAODNAAPS (aerosol innovations) for the SEA domain with a separate distributions for land and ocean retrievals. Recall that the innovation is simply the observed AOD minus the AOD derived from NAAPS reanalysis. Only retrievals where MODIS, CALIOP, and NAAPS all detect non-zero AOD are

- 20 included. CALIOP was used to filter cirrus contamination and only include cloud-cleared AOD scenes. Ocean AOD innovations average 0.02 for MODIS\_NAAPS and \_0.05 for CALIOP\_NAAPS. Over ocean, the differences (MODIS\_CALIOP) between the observed AOD values average 0.08, much larger than the differences when compared with NAAPS, indicating that the NAAPS bias is small. Over land, MODIS-NAAPS innovation has a mean of 0.05 compared to 0.07 for CALIOP\_NAAPS. Land innovations have a wider distribution than ocean innovations for both MODIS\_NAAPS and
- 25 CALIOP-NAAPS. Over land, the MODIS-NAAPS innovations have a variance of 0.04 while the CALIOP-NAAPS innovations have a variance of 0.2. Over ocean, these variances are 0.005 and 0.02 for MODIS-NAAPS and CALIOP-NAAPS, respectively.

The innovations for MODIS\_NAAPS are smaller than those for CALIOP-NAAPS, Campbell et al. (2012) states that CALIOP is known to have a low bias. Overall, the AOD biases are small, with NAAPS in good agreement with the MODIS

30 and CALIOP observations. NAAPS is understandably in slightly better agreement with MODIS, as the MODIS AOD is assimilated into the NAAPS reanalysis. However, MODIS assimilation occurs less frequently in SEA than in other regions

17

| Deleted: s                |
|---------------------------|
| Deleted: compute          |
| Deleted: ; thresholds are |
|                           |
| Deleted: Section          |
| Deleted: the              |

| Deleted: Figure |  |
|-----------------|--|
| Deleted: the    |  |
|                 |  |

**Deleted: aerosol optical depths**

| eleted: later           |
|-------------------------|
| eleted: Section         |
| eleted: Figure          |
| ormatted: Font:Not Bold |
| eleted: –               |
| ormatted: Font:Not Bold |
| eleted: –               |
| ormatted: Font:Not Bold |
| eleted: –               |
| ormatted: Font:Not Bold |
| eleted: –               |
| eleted: –               |
| eleted: comparing       |
| eleted: ocean AOD       |
| eleted: –               |
| eleted: –               |
| eleted: -               |
| eleted: –               |
| eleted: ;               |
| eleted: generally       |
| eleted: that            |
|                         |

due to persistent cirrus cloud cover. For CALIOP, the retrieval requires assumptions regarding the aerosol lidar ratio which can introduce systematic biases (Campbell et al., 2012).

The CCARA dataset provides vertically-distributed NAAPS and CALIOP aerosol extinctions, as discussed in Sect. 2.3.2. We compare layer extinctions in Fig. 5, which present biases and absolute values of aerosol extinction for PBL and MSL layers. For each layer, mean aerosol extinction and mean extinction bias are shown. The bias is the observed layer extinction

minus the NAAPS reanalysis layer extinction.

5

The top panels of Fig. 5 show the absolute extinctions (left) and extinction biases (right) for NAAPS and CALIOP MSL layers. Bias is defined as model extinction minus CALIOP observed extinction. Over both ocean and land, NAAPS overestimates CALIOP extinction at higher altitudes, and underestimates CALIOP extinction at lower altitudes. This is

10 consistent with the AOD bias results in Fig. 4. The extinction bias switches from positive to negative at 3.5 km over land and 1.5 km over ocean. The bottom panels of Fig. 5 show extinctions and extinction biases for the PBL layers, which are a finer vertical resolution near the surface compared to MSL layers. The extinction biases are exaggerated for the finer vertical bins, demonstrating NAAPS's underestimation of CALIOP layer bias over land. The PBL layer closest to the surface reports an extinction bias < 60.07 km-1 for land retrievals. In summary, we consider NAAPS aerosol reanalysis in reasonable
 15 agreement with MODIS and CALIOP observations. However, NAAPS does demonstrate lower aerosol particle concentrations in the lower free troposphere compared to CALIOP.

**3.1.2 Aerosol anomaly threshold**

Aerosol anomalies are defined by calculating the geometric mean ( $\mu_{AOD}$ ) and geometric standard deviation ( $\sigma_{AOD}$ ) of NAAPS fine mode aerosol optical thickness for each sub-region and each season (boreal summer (JJASON) and boreal

20 winter (DJFMAM)) during all ten years of NAAPS reanalysis collocated into the CCARA dataset. The NAAPS fine mode, AOD can be thought of as encompassing AOD contributions from both anthropogenic and smoke aerosol species. NAAPS AOD are typically log-normally distributed (e.g., Lynch et al., 2016), therefore geometric  $\mu_{AOD}$  and geometric  $\sigma_{AOD}$  are calculated.

We define a positive aerosol anomaly, or polluted event, when the AOD is greater than an upper threshold, ( $\mu_{AOD} \sigma_{AOD}$ ). A

25 negative aerosol anomaly, or clean event, is when the AOD is less than a lower threshold, (μAOD φAOD). Winter and summer upper and lower AOD thresholds for each region are shown in Table 1, along with the mean AOD values. Recall that to define ocean aerosol anomalies, the region is split into four large grids, which we will distinguish by referring to as southwest, northwest, southeast, and northeast ocean grids. Note the seasonal differences in each region and recall that Indochina experiences a biomass burning season during winter months while Java, Sumatra, and Borneo experience a

30 biomass burning season during summer months. These burning seasons are reflected in our AOD thresholds. Summer upper AOD thresholds for Java, Borneo, and Sumatra are much greater than their winter upper thresholds. Over Indochina, the

18

| Deleted: into the AOD retrieval                     |
|-----------------------------------------------------|
| Deleted: Section                                    |
| Deleted: Figure                                     |
|                                                     |
| Deleted: Biases separate land and ocean retrievals. |
| Deleted: Figure                                     |

| Deleted:             | [ [1] |
|----------------------|-------|
| Deleted: full-column |       |
| Deleted: Figure      |       |
| Deleted: Figure      |       |

| Formatted: Font:Not Bold |  |
|--------------------------|--|
| Deleted: -               |  |

| Deleted: ,           |
|----------------------|
| Deleted: ,           |
| Deleted: ,           |
| Deleted:             |
| Deleted: full-column |
| Deleted: Because     |
| Deleted:             |
| Deleted:             |
| Deleted:             |
| Deleted:             |
| Deleted: median      |
|                      |
| Deleted: smoky       |

winter upper AOD threshold over double that in the summer. Mean AOD values for each region and season are mapped in Fig. 6.

|                     | Summer                                    | Summer                  | Summer                                      | Winter                                    | Winter      | Winter                                      |
|---------------------|-------------------------------------------|-------------------------|---------------------------------------------|-------------------------------------------|-------------|---------------------------------------------|
|                     | $\underline{\mu_{AOD}}{\div}\sigma_{AOD}$ | µ AOD | $\underline{\mu_{AOD}}{\times}\sigma_{AOD}$ | $\underline{\mu_{AOD}} \div \sigma_{AOD}$ | µaod | $\underline{\mu_{AOD}}{\times}\sigma_{AOD}$ |
| Land - Indochina    | 0.06                               | 0.12             | 0.26                                 | 0.10                               | 0.23 | 0.57                                 |
| Land - Philippines  | 0.04                               | 0.08             | 0.17                                 | 0.03                               | 0.06 | 0.13                                 |
| Land - E. Indonesia | 0.04                               | 0.08             | 0.15                                 | 0.03                               | 0.06 | 0.11                                 |
| Land - Java         | 0.11                               | 0.19             | 0.32                                 | 0.08                               | 0.15 | 0.28                                 |
| Land - Borneo       | 0.06                               | 0.13             | 0.30                                 | 0.05                               | 0.09 | 0.15                                 |
| Land - Sumatra      | 0.06                               | 0.15             | 0.34                                 | 0.07                               | 0.13 | 0.24                                 |
| Ocean - northwest   | 0.04                               | 0.09             | 0.20                                 | 0.07                               | 0.16 | 0.33                                 |
| Ocean - northeast   | 0.02                               | 0.06             | 0.15                                 | 0.02                               | 0.06 | 0.17                                 |
| Ocean - southeast   | 0.03                               | 0.07             | 0.15                                 | 0.02                               | 0.05 | 0.11                                 |
| Ocean - southwest   | 0.03                               | 0.07             | 0.18                                 | 0.02                               | 0.05 | 0.11                                 |

Table 1: Lower fine mode AOD thresholds, median AOD, and upper AOD thresholds, shown for winter and summer for each of the six land regions and four ocean grids. Thresholds were found by computing the geometric mean ( $\mu_{AOD}$ ) and

geometric standard deviation ( $\sigma_{AOD}$ ). The lower threshold is  $\mu_{AOD}$ ;  $\sigma_{AOD_{av}}$  any AOD below this is considered *clean*. The upper threshold is  $\mu_{AOD} \times \sigma_{AOD}$  any AOD above this is *polluted*.

**3.1.3 Baseline MODIS effective radius**

In this section, we show the average (geometric mean or  $\mu_{z}$ ) MODIS liquid effective radius for each region and season. We also filter effective radius by three cloud height regimes, which are introduced in Sect. 2.4. These regimes are cloud top. 10 heights between 1-3 km, 3-4.5 km, and 4.5-6 km.

5

An effective radius population with no aerosol anomaly filtering is referred to as a 'baseline' population. Populations of baseline rent are compiled for each region and season for the ten-year dataset and the geometric mean of rent is computed. MODIS liquid refretrievals are included only if CALIOP detects no cirrus above the liquid cloud top. Pixels classified as partly cloudy (PCL) by the MODIS cloud mask algorithm (MOD35) are not included to reduce uncertainty in the effective radius results.

15

Maps of baseline geometric mean reff are shown in Fig. 7, These maps are shown for each season and cloud height regime. In each map, three black bolded numbers accompany each region. The first signifies the population count and the second preceded by an s is the geometric standard deviation of the baseline rent population. The s-number represents the "spread" of the baseline  $r_{\rm eff}$  population. The third number is the preceded by an *m* is the median  $r_{\rm eff}$ . As expected, average ocean  $r_{\rm eff}$

20 values are larger than land values as oceans regions are typically cleaner environments. Further, reff is generally larger on the

19

| Deleted: ion          |   |
|-----------------------|---|
| Deleted: -            | ٦ |
| Deleted: -            | 7 |
| Deleted: -            |   |
| Deleted: concatenated |   |
|                       | _ |

fringes of the domain, compared to reff in the higher populated areas. This is especially true for the ocean grids over the West Pacific Ocean in the northeastern-most region of the SEA domain (see also Fig. 8c), where average reff commonly exceeds 18 microns, compared to 8-10 microns over densely populated peninsular SEA and Java. Average reff values also increase with increasing height, likely resulting from the growth of droplets in deeper clouds. Evidence of major burning

5 seasons in Fig. 7 is seen over Indochina where the mean ref for all cloud heights is much smaller during burning active winter compared to summer months. Similar small reff over Java is seen during summer months. These maps suggest signs of a FAIE. Areas that are known to be polluted/clean experience smaller/larger effective radius retrievals. We now investigate if the mean reff values are statistically smaller/larger during extreme polluted/clean events in Sect. 3.2.

**3.2 FAIE occurrence**

- 10 Populations of MODIS liquid reff are compiled from the CCARA dataset filtering each region and season. We can filter these populations by aerosol thresholds using NAAPS aerosol reanalysis, allowing us to compare clean and polluted reff populations to baseline reff, which we consider the 'normal' reff. The reff can be filtered by other fields in the CCARA dataset such as CALIOP cloud top height. From the analysis in this section, we establish significant evidence for the FAIE. Distributions of these populations are shown in Sect. 3.2.1, where clean and polluted reff are compared to baseline reff. In
- 15 Sect. 3.2.2, we map the mean values of these distributions and investigate whether differences in distributions have statistical significance. Two-sample t-tests are conducted on the data to determine numerically whether means of clean/polluted reff are significantly larger/smaller than the baseline refi. Significant FAIE is detected if the two-sample t-test provides a p-value < 0.01.
- It is important to distinguish the direction of the FAIE. A polluted FAIE signifies that reff is smaller, relative to the region's 20
- baseline reff. A clean FAIE signal signifies that reff is larger than the baseline reff.

**3.2.1 Populations of reff for baseline, polluted, and clean cases**

Distributions of MODIS retrieved refi were compiled separating each region and season, and cloud height regime. For conciseness, select examples of those distributions are shown in Fig. 8. The figure shows normalized distributions of baseline, polluted, and clean reff of clouds between 1,3 km for six land regions: Indochina and the Philippines in winter and

- 25 Borneo, Java, East Indonesia, and Sumatra in summer and as highlighted by the coloured regions in the lower right of Fig. & Distributions for the West Pacific and Indian Oceans are shown as well. The black lines show the baseline distributions of reff, i.e. reff without aerosol anomaly. Note that the baseline distributions vary widely in shape between each region. Figs. 8a, (Indochina winter) and 8c (Java summer) have baseline reff with a very small mode and small variance compared to Figs. 8b. 8c, 8d, 8f, and 8h, whose baselines are more widely distributed.
- 30 During horeal winter, drier conditions and enhanced biomass burning are present in the northern half of our study domain. Indochina distributions of  $r_{eff}$  are very similar between baseline, polluted, and clean retrievals (Fig. 8a). This suggests that

20

| Deleted: –       |          |  |
|------------------|----------|--|
| Deleted: outhe   | ast Asia |  |
| Deleted: Figur   | e        |  |
| Deleted: 6       |          |  |
| Deleted: occur   |          |  |
| Deleted: Section | on       |  |

| Deleted: -                                              |    |
|---------------------------------------------------------|----|
| Deleted: - in cases of excess pollution                 |    |
| Deleted: in clean seemes that are leaking series | -1 |

| 4  | Deleted: ure 7     |
|----|--------------------|
| 4  | Deleted: -         |
| 4  | Deleted: Figure    |
| A  | Deleted: 7         |
| Å  | Deleted: any       |
| 4  | Deleted: filtering |
| 4  | Deleted: Panels    |
| -1 | Deleted: (a)       |
| -  | Deleted: (e)       |
| 1  | Deleted: panels    |
| ٦  | Deleted: (         |
| ٦  | Deleted: )         |
| 1  | Deleted: B         |

changes in aerosol concentration are not influencing significant change in the effective radius retrievals for the region, even during the burning season. This is unsurprising owing to high levels of biofuel, agricultural waste and industrial emissions in this region (e.g., Reid et al., 2013). In short, background conditions may already saturate the CCN population. In comparison, the Philippines distributions (Fig. 8b) are different between the baseline, polluted, and clean retrievals during

5 winter months. Polluted Philippines retrievals are smaller than the baseline; clean Philippines retrievals are larger than the baseline. In contrast to Indochina, the Philippines are more sensitive to FAIE during winter months with the fine mode aerosol concentration impacting the MODIS effective radius retrievals over this region. Finally, the most dramatic sensitivity, can be seen for the West Pacific boreal winter (8c), the most remote of the regions examined.

During boreal summer, the monsoon flips to more wet conditions in the north and drier, more fire-prone conditions in the 10 south. In Fig. 8d, Borneo behaves as would be expected owing to its relatively high population, little sensitivity in the

baseline and more polluted conditions, but with some residual sensitivity for the "cleanest" of conditions. Such cleaner conditions do not exist on the highly populated island of Java, with considerably less sensitivity to modeled perturbations in AOD. Similar findings were found for Sumatra (Fig. 8g). Some of the largest spreads in reff for boreal summer were found in the more remote areas, such as Eastern Indonesia over land (Fig. 8f) or the Jndian ocean – a frequent receptor for pollution
 from Java (Fig. 8h). Similar to the West Pacific winter in Fig. 8c, the Indian ocean experiences sensitivity during scenes of

anomalous aerosol.

While the examples in Fig. 8 hint at signals of clean and polluted FAIE signals, we must consider the statistical significance of these distributions in the comparisons. The next section will systematically explore clean and polluted FAIE signals for each region in SEA and investigate signals of the FAIE where statistical testing has deemed the FAIE significant.

**20 3.2.2 Mapped FAIE for winter and summer**

In this section, we compute geometric means of  $r_{eff}$  for baseline, polluted, and clean retrievals for each region, season, and cloud height regime. We use a t-test to determine whether the geometric means of those distributions are significantly different from one another to yield a FAIE signal. The significance test is designed such that we can test specifically if the clean  $r_{eff}$  is *larger* than the baseline  $r_{eff}$  (a clean FAIE) and if the polluted  $r_{eff}$  is *smaller* than the baseline  $r_{eff}$  (polluted FAIE).

 $25 \quad \text{The magnitude of the FAIE is the displacement of the anomalous (clean or polluted) } r_{\text{eff}} \text{ from the baseline } r_{\text{eff}}.$

clean FAIE = clean  $r_{eff}$  – baseline  $r_{eff} \ge 0$

polluted FAIE = polluted  $r_{eff}$  – baseline  $r_{eff} \! \leq \! 0$

This definition is defined so the clean FAIE is positive in sign and the polluted FAIE is negative. Fig. 9, shows maps of average reff values for baseline populations, as well as maps of polluted and clean FAIE signals during summer months.

30 Winter months are shown in Fig, 10. The rows in this figure signify different cloud top height regimes; the columns show the baseline reff, polluted FAIE signal, and clean FAIE signal. The cloud height regimes signified by the rows in Figs. 9, and 10, correlate from bottom to top: 1, 3 km, 3, 4.5 km, and 4.5.6 km. The FAIE signal maps are only shaded if the FAIE is deemed

| 1  | 1 |
|----|---|
| 1. | Т |
| _  | • |

| ~ |                                          |
|---|------------------------------------------|
| U | Deleted: the regions                     |
|   |                                          |
| 6 | Deleted:, also in a drier winter period, |
|   |                                          |
| ſ | Deleted: the                             |
|   |                                          |
| ( | Deleted: is in                           |
| ſ | Deleted: Moving to the                   |
| ſ | Deleted: to                              |
| [ | Deleted:                                 |
| 6 | Deleted: to                              |
| G | Deleted: Panel D                         |

| Deleted: Figure |  |
|-----------------|--|
| Deleted: 7      |  |

| Å | Deleted: Figure  |
|---|------------------|
| 4 | Deleted: 8       |
| Å | Deleted: ure     |
| A | Deleted: 9       |
| Å | Deleted: Figures |
| 4 | Deleted: 8       |
| 1 | Deleted: 9       |
| 4 | Deleted: –       |
| - | Deleted: –       |
| 1 | Deleted: -       |

statistically significant (p < 0.01). Note that the same region may yield a significant clean FAIEa but not a significant polluted FAIE, or vice versa.

During winter months shown in Fig. 10, for clouds nearest to the surface (between 1-3 km) there is a significant clean and polluted FAIE (up to 6 microns in magnitude) for the West Pacific Ocean region, just east of the Philippines. This suggests

5 that under heavily polluted environments, the average effective radius is up to 6 microns smaller relative to the effective radius under every-day scenes. Under extremely clean scenes, the average effective radius can be over 5 microns larger than the bulk average. Even for high clouds at 4.5 for km, a polluted and clean FAIE signal is found for some of the West Pacific grids. A polluted and clean FAIE is also present for the Indian Ocean during winter (Fig. 10) and summer (Fig. 9). It appears maritime regions such as the West Pacific and Indian Oceans are some of the most susceptible to changes in reff during

**10 anomalous aerosol amounts.**

A very weak (< 1 micron in magnitude) clean and polluted FAIE signal exists for low clouds over Indochina. These FAIE signals are weak as the environment is likely almost always saturated with fine mode aerosol, resulting in additional sources having limited impact on the water cloud droplet size. Comparing Indochina in Fig. 9, to Fig. 8a, note that while the normalized histograms of clean and polluted  $r_{\rm eff}$  are almost identical, the slight difference in the right-side tails of this

- 15 distribution, in addition to the high number of counts, yields a p-value small enough to consider this difference significant. The example of wintertime Indochina is consistent with the notion that there is a physical upper limit to cloud-aerosol interactions (Painemal and Zuidema, 2013). A slightly stronger FAIE over Indochina does appear for higher altitude clouds, though the average reff for all (baseline) clouds over Indochina stays quite small and does not exceed 12 microns even for the tallest clouds.
- 20 The black bolded numbers in Figs. 9 and 10 signify the count of the cloud populations for each region in. The purpose of showing retrieval counts is to indicate whether the lack of a FAIE signal is due to low cloud counts or an insignificant change in the average anomalous reff.

Over some West Pacific peripheral ocean regions in Fig. 10, there is a lack of polluted liquid  $r_{eff}$  retrievals for higher clouds (3.6 km). One reason for this is that the region is regularly covered in cirrus, reducing the likelihood of a liquid retrieval

- 25 from MODIS. Additionally, while significant smoke and pollution can reach this region, it is somewhat rare and compounded by how we define aerosol anomaly. Our aerosol thresholds were defined by determining whether a certain AOD amount was anomalous relative to all clouds in that region during that season, *not* answering whether the AOD was anomalous to specific cloud top heights in that region during that season. Further, recall that ocean aerosol anomalies are determined for larger ocean grids, not the small 10x10-deg grids in Figs. 9 and 10. For these reasons, it follows that some
- 30 easternmost West Pacific polluted FAIE grids in Fig. 10 indicate a "0" count for taller cloud height regimes because any anomalously polluted clouds in that region are mostly likely to exist in the lowest regime closest to the surface. During summer months shown in Fig. 9, a strong polluted FAIE appears across the SEA domain. Significant clean FAIE signals are present as well. Peripheral ocean regions in the West Pacific (upper right in maps) and Indian Ocean (lower left in

maps) experience a strong polluted FAIE, with clouds decreasing in average  $r_{eff}$  up to 6 microns for all cloud height regimes. 22

| -{ | Deleted: Figure     |
|----|---------------------|
| -{ | Deleted: 9          |
| `( | Deleted: -          |
| 1  | Deleted: retrievals |
|    |                     |
| {  | Deleted: -          |
| -{ | Deleted: Eastern    |

| Deleted: Figure                |  |
|--------------------------------|--|
| Deleted: 8                     |  |
| Deleted: Panel (a) in Figure 7 |  |

| Deleted: | Figure |
|----------|--------|
| Deleted: | 9      |
| Deleted: | -      |

| Deleted: | Figures 8 and 9 |  |
|----------|-----------------|--|
|          |                 |  |

It is clear that the burning season of Sumatra, Java, and Borneo has a great effect on cloud droplet size over ocean regions, with far-reaching effects into the West Pacific. It is likely that the nearly constant presence of anthropogenic aerosols from Indochina have an effect on West Pacific effective radius retrievals as well. The effect is emphasized during winter, which is Indochina's biomass burning season.

5 There is a complete absence of polluted FAIE signals over Java, Sumatra, and Borneo for 1,2 km clouds. These land regions have large populations, significant biofuel usage, and undergo burning seasons during these months. Java yields no polluted FAIE signal for *any* cloud heights. The mean reff values in baseline and clean cases over Java are relatively low at 13 microns, implying that these regions are saturated with aerosol particles. We infer that relative changes in the aerosol loading having minimal effect on cloud droplet sizes for low clouds over regions of biomass burning.

**10 4 Discussion and conclusions**

Southeast Asia has long been thought of as particularly vulnerable to climate change. High population density and proximity to remote ocean regions highlights the complexity of observing aerosol indirect effects in this region. In this analysis, we have shown that the effects of anthropogenic and smoke aerosols on cloud properties are detectable using the ten-year collocated Curtain Cloud-Aerosol Regional A-Train (CCARA) dataset, which includes data from MODIS and CALIOP and

- 15 model reanalysis from the NAAPS. The dataset uses aerosol reanalysis from NAAPS paired with concurrent satellite cloud observations to show significant signs of a first aerosol indirect effect or FAIE. This effect occurs when the mean effective radius is larger in anomalously clean cases and smaller in anomalously polluted cases when comparing to the overall (baseline) average reff. We call these effects the clean FAIE and polluted FAIE signals, respectively. FAIE signals are significant only if the reff populations of anomalous and clean scenes pass a significance t-test and yield p-values < 0.01.
- Sect. 3.2.2 shows that clean and polluted FAIE signals are quite common throughout the SEA domain during both winter and summer. Disregarding reff in scenes of aerosol anomaly, baseline average reff can vary greatly between regions. Biomass burning seasons are reflected in the baseline reff, as distributions of reff are often condensed to very small values over land, as shown in examples in Fig. 8, Furthermore, burning seasons cause saturation in CCN for low clouds (1-3 km) over land. In these regions, changes in pollution have a limited impact on observed effective radius.
- 25 Smoke burning and anthropogenic aerosols have long-reaching effects on cloud properties in SEA, particularly in maritime regions. Figs. 9 and 10 show that peripheral ocean regions are often the most affected by influxes of anthropogenic aerosol; this is seen in winter and summer months. Ocean regions have varying sensitivity to anomalous AOD based on biomass burning season. The region is especially sensitive to the FAIE even in areas that are thought of as remote or pristine. There is often a weak or non-existent FAIE over regions that are saturated with aerosol, i.e. low clouds (cloud top height 133).
- 30 km) over polluted land areas undergoing a biomass burning season. This is seen for Sumatra, Java, and Borneo during summer months and for Indochina during winter months. This result signifies that under high pollution events in already polluted regions, changes in aerosol amount have little to no effect on cloud microphysics. These same regions do sometimes

[revised manuscript text omitted]

**Regions for computing AOT anomaly**